# Improving LLM-based Global Optimization with Search Space Partitioning

**Andrej Schwanke**[1][*], **Lyubomir Ivanov**[1][*], **David Salinas**[2],
**Fabio Ferreira**[2,1], **Aaron Klein**[2], **Frank Hutter**[3,2,1] **& Arber Zela**[1,2][*]

[1]University of Freiburg   [2]ELLIS Institute Tübingen   [3]Prior Labs

## Abstract

Large Language Models (LLMs) have recently emerged as effective surrogate models and candidate generators within global optimization frameworks for expensive blackbox functions. Despite promising results, LLM-based methods often struggle in high-dimensional search spaces or when lacking domain-specific priors, leading to sparse or uninformative suggestions. To overcome these limitations, we propose HOLLM, a novel global optimization algorithm that enhances LLM-driven sampling by partitioning the search space into promising subregions. Each subregion acts as a "meta-arm" selected via a bandit-inspired scoring mechanism that effectively balances exploration and exploitation. Within each selected subregion, an LLM then proposes high-quality candidate points, without any explicit domain knowledge. Empirical evaluation on standard optimization benchmarks shows that HOLLM consistently matches or surpasses leading global optimization methods, while substantially outperforming global LLM-based sampling strategies.

## 1 Introduction and Motivation

Global optimization (Jones et al., 1998; Rios & Sahinidis, 2013) (also known as gradient-free or zeroth-order optimization) of blackbox functions, where the only information provided to the optimizer is the function value, is a fundamental challenge across numerous domains including hyperparameter tuning (Snoek et al., 2012; Turner et al., 2021), policy search (Calandra et al., 2016), molecular design and chemical engineering (Langer & Tirrell, 2004; Hernández-Lobato et al., 2017), just to name a few. Methods such as Bayesian optimization (Shahriari et al., 2016; Garnett, 2023) and evolutionary algorithms (Hansen, 2016) have been a standard and effective choice across various applications. However, they typically require assumptions regarding the underlying objective function's nature, which consecutively affect algorithmic design choices.

At the same time, recent advances in Large Language Models (LLMs) have demonstrated remarkable capabilities in generative modelling and reasoning (Brown et al., 2020; OpenAI, 2023; Touvron et al., 2023), suggesting their potential usage for optimization tasks as well (Song et al., 2024). Efforts in integrating LLMs within blackbox optimization algorithms as surrogate models or as candidate samplers have already shown encouraging results (Yang et al., 2024; Liu et al., 2024; Zhang et al., 2023; Aglietti et al., 2025; Agarwal et al., 2025; Kristiadi et al., 2024). Yet these methods typically rely on carefully engineered, domain-specific prompts, and in higher dimensions and complex search spaces the LLM's suggestions tend to scatter sparsely, covering only a fraction of the domain (Krishnamurthy et al., 2024).

As a motivating example, we investigated the capabilities of LLMs to simulate uniform sampling from a unit hypercube. In Figure 1a we show 80 samples drawn from the unit square $[0, 1]^2$, comparing uniform sampling (blue) with Gemini-1.5's (Reid et al., 2024) attempt at simulating uniform sampling using the prompt provided in Listing 1 in Appendix I (green points), and Gemini-1.5 performing uniform sampling with 5 samples per smaller subregion, using the same prompt (red points). We can clearly notice that even in 2D the LLM demonstrates high bias when sampling, therefore failing to appropriately fill the space as it was tasked to, whilst partitioning the space and prompting the LLM 16 times yields a more faithful simulation. Another illustrative example is shown in Figure 1b, where

---

[*]Equal contribution. Email to: `andrejschwanke19@gmail.com`, `arber.zela@tue.ellis.eu`

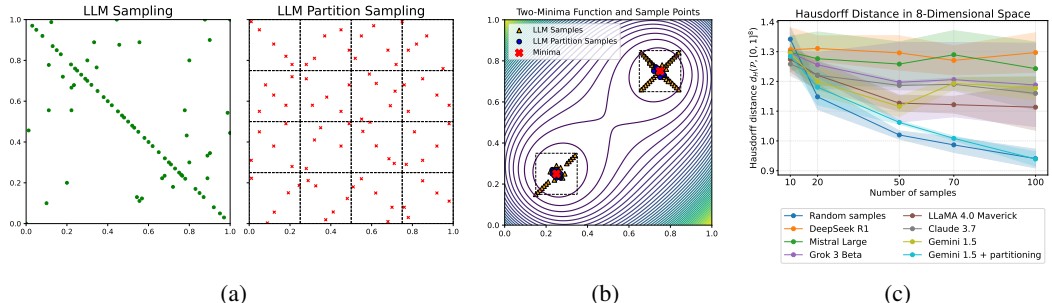

(a)                                       (b)                                       (c)

Figure 1: (*a*) 80 samples in $[0,1]^2$: Gemini-1.5 simulating uniform sampling (green), and with region-wise partitioning (red) using the prompt in Listing 1. (*b*) Gemini-1.5 prompted (see Listing 2) to generate 80 samples around the 2 minima (red crosses) globally (triangles) and withing the two bounding boxes (circles). (*c*) Hausdorff distance $d_H(\mathcal{P}, [0,1]^8)$ for uniform vs. LLM-simulated sampling in the 8-D hypercube.

we prompt Gemini-1.5 (using the prompt shown in 2) to sample close to the two global minima (red stars) of a quadratic function, given the input space boundaries. We can clearly notice the higher sampling bias when the input space is $[0,1]^2$ instead of the smaller regions denoted via the dashed bounding boxes. Finally, in Figure 1c, we compute the Hausdorff distance, $d_H(\mathcal{P}, [0,1]^8)$, between the set of $N \in \{10, 20, 50, 70, 100\}$ sampled points $\mathcal{P}$ and the 8-dimensional unit hypercube $[0,1]^8$. The blue curve indicates the values for standard uniform sampling and the other ones performed by various non-agentic LLMs. Similarly as in the 2D case, partitioning the hypercube into 32 regions and sampling within each (Gemini 1.5 + partitioning) notably improves the spatial coverage, enabling the LLM to more closely approximate uniform sampling.

In this paper, we introduce *Hierarchical Optimization with Large Language Models (HOLLM)*, a novel blackbox optimization method that leverages adaptive spatial partitioning to guide LLM-based sampling. HOLLM iteratively builds a KD-tree on existing evaluation data, creating adaptive local partitions whose granularity evolves with sampling density. Each subregion is assigned a bandit-inspired utility score, balancing exploitation (regions with promising observed values) and exploration (geometrically large or statistically uncertain regions). Subregions are selected stochastically according to these scores, and LLMs then generate localized candidate proposals within the chosen regions. As LLMs trained on optimization literature and scientific data encode a valuable *meta-prior* about typical function behavior (e.g., local unimodality), we effectively harness this prior without assuming a fixed parametric surrogate (e.g., Gaussian Process). Furthermore, restricting candidate generation to smaller, lower-dimensional subregions significantly reduces LLM sampling difficulty compared to global high-dimensional sampling. We note that spatial partitioning heuristics have already proven effective in continuum-armed bandits (Munos, 2011; Bubeck et al., 2011; Valko et al., 2013; Grill et al., 2015), Trust Region Bayesian optimization (Eriksson et al., 2019; Daulton et al., 2022), and Monte Carlo Tree Search (Kim et al., 2020; Wang et al., 2020; Yang et al., 2021). The key contribution of this work is the integration of these partitioning ideas to substantially improve LLM-driven global optimization performance.

Empirical evaluations on continuous and discrete benchmark functions, including hyperparameter optimization and real world tasks, demonstrate that HOLLM effectively balances exploration and exploitation, matching or outperforming state-of-the-art methods, including established Bayesian optimization variants and Trust Region algorithms, particularly in scenarios requiring efficient navigation of complex landscapes. Furthermore, compared to approaches that prompt the LLM to propose candidates globally, HOLLM achieves considerable gains by focusing LLM suggestions locally. We provide the implementation of our algorithm in the following repository: `https://github.com/arberzela/mohollm.git`.

## 2 BACKGROUND AND RELATED WORK

We consider the problem of maximizing a blackbox function $f : \mathcal{X} \to \mathbb{R}$ where $\mathcal{X}$ is a compact domain. The objective is to find $x^* = \arg\max_{x \in \mathcal{X}} f(x)$ through a sequence of function evaluations. In this blackbox setting, we do not have access to gradients or other properties of $f$, and can only

observe function values at queried points. The performance of optimization algorithms in this context can also be measured using *simple regret* or *cumulative regret*. For a sequence of evaluated points $x_1, x_2, \ldots, x_t$, the simple regret after $t$ iterations is defined as: $r_t = f(x^*) - \max_{i \in \{1, \ldots, t\}} f(x_i)$, while the cumulative regret is: $R_t = \sum_{i=1}^{t} (f(x^*) - f(x_i))$.

**Bayesian Optimization.** Bayesian Optimization (BO) (Garnett, 2023; Frazier, 2018; Shahriari et al., 2016) is a well-established framework for optimizing expensive blackbox functions by maintaining a probabilistic surrogate (typically a Gaussian Process (Rasmussen & Williams, 2006)) to guide evaluations and optimizing an acquisition function (e.g. Expected Improvement (Zhan & Xing, 2020)) in order to balance exploration and exploitation and efficiently search the space. Extensions like TuRBO (Eriksson et al., 2019; Daulton et al., 2022) address high-dimensional settings by maintaining multiple trust regions, which are dynamically resized based on optimization progress, enabling scalable and focused exploration around promising evaluations via local GPs.

**Multi-Armed Bandits and Hierarchical Optimization Algorithms.** Multi-Armed Bandits (MABs) (Slivkins, 2019) deal with the problem of sequential decision-making under the exploration-exploitation dilemma. In the basic setting, a MAB algorithm repeatedly selects among a fixed (also infinite) number of arms or actions, each with an unknown reward distribution, aiming to minimize the cumulative regret. In the global optimization setting, the arms are the points that lie in the input space $\mathcal{X}$ and at each iteration $t$, an arm $x_t \in \mathcal{X}$ is pulled and the regret is computed by evaluating the function $f(x_t)$ (Grill et al., 2015). Several MAB algorithms leverage hierarchical space partitioning (Kleinberg et al., 2008). Most notably, HOO (Bubeck et al., 2011) constructs a hierarchical partitioning of the search space using $n$-ary trees and at each step, an unexplored region (tree leaves) is selected based on upper confidence bounds (UCB) (Auer et al., 2002; Kocsis & Szepesvari, 2006) and $f$ is evaluated at a point uniformly sampled inside the selected region. Building on HOO, extensions include parallel versions (Grill et al., 2015), optimization without explicit smoothness knowledge (Munos, 2011) or under noisy observations (Valko et al., 2013), and adaptive trees (Bull, 2013) including Monte Carlo bandits (Wang et al., 2020; Yang et al., 2021). Most of these methods come with theoretical guarantees on regret bounds that depend on the dimensionality and smoothness properties of the objective function.

**Large Language Models for Blackbox Optimization.** Recent work has increasingly explored integrating LLMs into blackbox optimization workflows. Some approaches prompt LLMs directly to generate candidate solutions in natural language (Liu et al., 2024; Yang et al., 2024; Agarwal et al., 2025; Zhang et al., 2023), use them to estimate uncertainty (Ramos et al., 2023), extract features (Kristiadi et al., 2024), or even design novel acquisition functions (Aglietti et al., 2025). Others replace traditional surrogate models with LLMs to predict function values and guide search in applications such as hyperparameter tuning (Liu et al., 2024) and molecular design (Ramos et al., 2023). However, these methods often rely on carefully engineered prompts containing domain-specific information (e.g., dataset statistics or problem descriptions), raising concerns about their robustness in domains where this information is not available. Recent work by Krishnamurthy et al. (2024) shows that, in simple MAB settings, LLMs struggle to explore effectively without significant prompt intervention, highlighting their limitations in decision-making.

Our algorithm builds upon these foundations in order to improve LLM-based blackbox optimization by integrating tree-based space partitioning, a UCB-inspired score function for balancing exploration and exploitation, and LLM-based candidate generation within locally promising regions.

## 3  HOLLM: HIERARCHICAL OPTIMIZATION WITH LLMS

In this section, we present the HOLLM algorithm for optimizing potentially noisy blackbox functions $f : \mathcal{X} \to \mathbb{R}$, which consists of 5 main steps: `Partition`, `Score`, `Select`, `Sample` and `Evaluate`. Given an initial set of $n_0$ evaluations $\mathcal{D}_{n_0} = \{(x_i, f(x_i))\}_{i=1}^{n_0}$, the algorithm iteratively calls each of these steps. It starts by adaptively partitioning the search space in a data-driven way, scores each of these regions to balance exploration-exploitation, selects the $M$ most promising regions based on their score, leverages LLMs to sample candidates within these regions, and finally evaluates the best candidates according to their predicted function value from the LLM. We provide an illustrative depiction of these steps in Figure 2. This approach allows the LLM to focus on promising smaller regions of the space while benefiting from the global partitioning strategy. We

provide the algorithm pseudocode in Algorithm 1 and a more detailed version in the Appendix A. In the following, we explain each step in detail.

## 3.1 PARTITION: ADAPTIVE DISCRETIZATION

Based on the motivating examples we presented in Section 1, we hypothesize that firstly identifying promising smaller regions in the input space $\mathcal{X}$ makes the LLM-based sampling more reliable compared to prompting the LLM to sample globally. To this end, we propose using an adaptive input space partitioning method based on the evaluated data at each iteration of the algorithm. In order to obtain disjoint space partitions that cover the entire space, we use k-dimensional trees (KD-trees), a space-partitioning data structure that recursively divides the space into half-spaces, so that we can efficiently compute the partitions in high dimensions ($\mathcal{O}(t \log(t))$ for a balanced tree where $t$ is the number of iterations), whereas for other methods, such as a Delaunay triangulation (Gramacy et al., 2022) and Voronoi diagram (Kim et al., 2020; Wycoff et al., 2024), this would become quickly impractical as the dimension $d$ increases. Each non-leaf node in a KD-tree represents a splitting hyperplane perpendicular to one of the coordinate axes, dividing the space into two parts. Points on the "left" side of this hyperplane are represented by the left subtree, and points on the "right" side are represented by the right subtree. Starting from the root node $\mathcal{X}_\varnothing = \mathcal{X}$, every internal node chooses a split dimension $s$ (the one with the largest variance among points in the node) and a split value $\delta$ (the mean across the selected dimension). This produces two child nodes

$$\mathcal{X}_{\text{left}} = \{x \in \mathcal{X} : x_s \leq \delta\}, \quad \mathcal{X}_{\text{right}} = \{x \in \mathcal{X} : x_s > \delta\},$$

whose union equals their parent and whose interiors are disjoint. After inserting $n$ sample points, the $K$ leaves $\{\mathcal{X}_l\}_{l=1}^K$ form a partition of $\mathcal{X}$ into axis-aligned hyperrectangles and contain information about the points evaluated within it, including their coordinates and function values. We denote the set of indices each leaf $\mathcal{X}_\ell$ holds as: $I_\ell = \{ i \leq t : x_i \in \mathcal{X}_\ell \}$, with sample size $n_\ell = |I_\ell| \leq m_t$. $m_t$ is the maximum number of points a leaf in the KD-tree can keep before splitting, parameterized by the number of iterations. At the start of round $t$, we optionally set $m_t = m_0 + \lceil \lambda \log(1 + t) \rceil$, where $m_0$ ($\lceil d/2 \rceil$ by default) is the initial leaf size and $\lambda$ (0 by default) is the growth parameter. The logarithmic growth of $m_t$ ensures that the partitions do not become too fine-grained quickly.

**An infinite-armed bandit view.** Conceptually, the KD-tree can be interpreted as a *data-driven discretizer* in infinite-armed bandits (Wang et al., 2008; Carpentier & Valko, 2015): its leaves form a coarse partition at the beginning of learning and refine only where information accumulates, mirroring the "zooming" phenomenon in continuum–armed bandits (Kleinberg et al., 2008; Wang et al., 2008; Munos, 2011; Bubeck et al., 2011; Slivkins, 2011; Valko et al., 2013; Bull, 2013; Carpentier & Valko, 2015; Grill et al., 2015). A key distinction, however, is that the entire tree is *re-fitted* at every round, adapting the partitions' boundaries based on the current data to avoid potential early convergence to local minima. This strategy is similar to Wang et al. (2020); Yang et al. (2021), where they observed that recomputing the partitioning every few iterations resulted in better empirical performance. Although partition boundaries may merge or shift, one can frame the procedure as operating on a *fixed, infinite* KD-tree whose internal nodes are activated and deactivated on the fly, as abstracted in adaptive-treed bandits (Bull, 2013; Slivkins, 2011). Following this paradigm, every axis-aligned hyper-rectangular partition, $\mathcal{X}_l \subset \mathcal{X}$, can be seen as a "meta-arm" (Slivkins, 2011). The reservoir of all such boxes is uncountable, hence discovering arms, rather than merely pulling them, becomes part of the learning problem. In this language, our algorithm may be viewed as an infinite-armed bandit strategy that *(i)* repeatedly draws a batch of candidate active input space partitions by re-fitting a KD-tree to the ever-growing set of evaluations, and *(ii)* allocates pulls among those boxes according to a score function (as described below).

## 3.2 SCORE: SYNTHESIS OF EXPLOITATION, GEOMETRY AND UNCERTAINTY

At every iteration $t$ the KD-tree yields a finite collection of leaves (partitions) $\{\mathcal{X}_{\ell,t}\}_{\ell=1}^{K_t}$. In order to decide where to spend our limited evaluation budget, we need to rank these leaves based on a scoring function (also called utility or acquisition function) that balances exploitation of good leaves and exploration of large regions that may hold good points and that are under-sampled.

**(i) Exploitation via the empirical *maximum*.** The exploitation term should *optimistically* reflect the best empirical evidence available for each region. Classical HOO algorithms use the sample mean as a

Figure 2: Overview of the HOLLM algorithm: starting from initial data $\mathcal{D}$, it iteratively performs `Partition`, `Score`, `Select`, `Sample` (via LLM), and `Evaluate` steps to balance exploration and exploitation. For the partitioning here, we utilized a KD-Tree where each axis is split based on the mean values. Each rectangle represents a partition defined by the tree leaves. The red stars represent the new sampled points from the LLM.

low-variance proxy for the local reward (Kleinberg et al., 2008; Wang et al., 2008; Bubeck et al., 2011; Valko et al., 2013). In global optimization, however, the objective may be highly *heteroscedastic*, where one exceptionally good point inside an otherwise mediocre box can be more informative than the entire distribution. We therefore let our exploitation statistic be the largest improvement ever observed in a region $\mathcal{X}_{\ell,t}$:

$$f_{\min}(t) = \min_{i \leq t} f(x_i), \quad Y_i = f(x_i) - f_{\min}(t) + \varepsilon, \quad \mu_{\ell,t} = \max_{i \in I_{\ell,t}} Y_i. \tag{1}$$

We subtract the current empirical minimum $f_{\min}(t)$ (since we are maximizing $f$) so the values become strictly non-negative and comparable across rounds [1]. Choosing a *max* rather than an average emphasizes regions that contain a good function value, a behavior also found in acquisition functions in Bayesian optimization (Garnett, 2023) and MCTS (Wang et al., 2020; Yang et al., 2021).

**(ii) Geometric exploration through *hypervolume*.** Let $[l_{\ell_1}, u_{\ell_1}] \times \cdots \times [l_{\ell_d}, u_{\ell_d}]$ be the axis-aligned hyperrectangle corresponding to leaf $\mathcal{X}_{\ell,t}$, where $l_{\ell_d}$ and $u_{\ell_d}$ are the low and upper axis values across dimension $d$ determined by the points in $\mathcal{X}_{\ell,t} = \{x_i \in \mathcal{X} : i \in I_{\ell,t}\}$. In order to assign a high exploration score to regions in the input space that are underexplored, we use the $d$-th root of the leaves' Euclidean volume $\mathrm{Vol}(\mathcal{X}_{\ell,t})$: $V_{\ell,t} = \left( \prod_{j=1}^{d} (u_{\ell_j} - l_{\ell_j}) \right)^{1/d}$, which is equivalent to the geometric mean of the side lengths of the hyperrectangle and is less sensitive to side lengths across single dimensions compared to the cell diameter. The $d$-th root scales $\mathrm{Vol}(\mathcal{X}_{\ell,t})$ so it has the same units as a *length*. Because axis-aligned boxes shrink anisotropically as the KD-tree refines, the $d$-th root removes the strong dependence on dimension and yields comparable numbers across $d$.

**(iii) Statistical exploration via a UCB–V term.** Even a tiny region may deserve further sampling if it contains a few samples with high variance. Let $\sigma_{\ell,t}^2$ be the empirical unbiased variance of the observed function values $\{Y_i\}_{i \in I_{\ell,t}}$ within region $\mathcal{X}_{\ell,t}$ at iteration $t$, and let $n_{\ell,t} = |I_{\ell,t}|$ be the number of samples in that cell. We adopt an exploration factor inspired by UCB-V (Upper Confidence Bound with Variance estimates) type algorithms (Audibert et al., 2009; Audibert & Bubeck, 2009; Wang et al., 2008; Mukherjee et al., 2018), and apply it to our dynamic KD-tree partitioning, reminiscing UCB-AIR for infinite-armed bandits where the number of arms increases at each iteration (Wang et al., 2008). More specifically, we score the region $\mathcal{X}_{\ell,t}$ with:

$$\mathcal{E}_{\ell,t} = \sqrt{\frac{2\,\sigma_{\ell,t}^2\,\max\big(0, \ln(t/(K_t n_{\ell,t}))\big)}{n_{\ell,t}}} + \frac{c \cdot \max\big(0, \ln(t/(K_t n_{\ell,t}))\big)}{n_{\ell,t}}. \tag{2}$$

Here, $K_t = |\{\mathcal{X}_{\ell,t}\}|$ is the current number of active leaves (partitions) in the KD-tree at iteration $t$, and $c$ is a positive constant [2]. The argument of the logarithm, $t/(K_t n_{\ell,t})$, compares the average number of samples per region $(t/K_t)$ to the samples $n_{\ell,t}$ in the specific region $\mathcal{X}_\ell$. This is a concentration term that focuses exploration on regions sampled less frequently than the current average. The $\max(0, \ln(\cdot))$ ensures the logarithmic term contributes non-negatively, effectively diminishing direct exploration incentive from this term for regions sampled more than average relative to $K_t$. Since the effective noise or function variability can vary significantly across regions, we scale this concentration

---

[1]The additive constant $\varepsilon$ prevents zero scores during the startup phase.

[2]$c$ is often related to the range of function values or is a tuning parameter. We set it to 1 since in the total score we weight the total exploration factor.

---

**Algorithm 1:** HIERARCHICAL OPTIMIZATION WITH LLMS (HOLLM)

---

**Data:** Initial data $\mathcal{D}$, budget $T$, batch size $b$, regions to sample from $M$, proposals per region $k$

1. **while** $t \leq T$ **do**
2.     Update temperature $\alpha_t$ (and optionally maximum leaf size $m_t$)
3.     Partition space by building KD-tree on $\mathcal{D}$, obtaining $K_t$ leaves $\{\mathcal{X}_{\ell,t}\}_{\ell=1}^{K_t}$     // Partition
4.     **for** *each leaf $\mathcal{X}_{\ell,t}$* **do**
5.         Compute $\mu_{\ell,t}$ (Eq. 1), $V_{\ell,t} = \text{Vol}(\mathcal{X}_{\ell,t})^{1/d}$ and $\mathcal{E}_{\ell,t}$ (Eq. 2)
6.         Normalize and compute total score $B_{\ell,t} = \bar{\mu}_{\ell,t} + \alpha_t \left( \beta_1 \bar{V}_{\ell,t} + \beta_2 \overline{\mathcal{E}}_{\ell,t} \right)$     // Score
7.     **end**
8.     Select $M$ leaves by sampling with probabilities $p_{\ell,t} \propto B_{\ell,t}$     // Select
9.     Generate $k$ candidates for each chosen leaf via LLM_GENERATE($\mathcal{D}, \mathcal{X}_{\ell,t}, k$)     // Sample
10.     Pick the top $b$ proposals by their LLM predicted scores
11.     Evaluate $f$ on them, add to $\mathcal{D}$, and set $t \leftarrow t + b$     // Evaluate
12. **end**
13. **return** *best* $(x, y) \in \mathcal{D}$

---

term inside the first summand with the empirical variance $\sigma_{\ell,t}^2$ of the corresponding region. The second summand is a correction term characteristic of Bernstein-style concentration bounds (Audibert et al., 2009; Maurer & Pontil, 2009). It helps to ensure that the exploration bonus is sufficiently large, particularly when $n_{\ell,t}$ is small or when the empirical variance $\sigma_{\ell,t}^2$ happens to be small or zero [3]. This makes the exploration strategy more robust for leaves with limited observations.

**Final composite score.** All components must live on a shared numeric scale; otherwise, whichever component happens to have the largest dynamic range would dominate the others and nullify the intended trade–off. After each rebuild, we normalize the scores to $[0, 1]$, preserving the intended relative weights even when the set of leaves changes drastically. The total score of each partition determined by the KD-tree partitioning is:

$$B_{\ell,t} = \bar{\mu}_{\ell,t} + \alpha_t \left( \beta_1 \bar{V}_{\ell,t} + \beta_2 \overline{\mathcal{E}}_{\ell,t} \right), \tag{3}$$

where $\bar{\mu}_{\ell,t}, \bar{V}_{\ell,t}, \overline{\mathcal{E}}_{\ell,t}$ are the min-max normalized scores, and $\beta_1, \beta_2$ are hyperparameters ($\beta_1 + \beta_2 = 1$ by default) weighting the geometric versus statistical exploration. The $\alpha_t$ multiplier is a total exploration weight following an annealing schedule (cosine in our experiments). In the early phase ($\alpha_t \approx \alpha_{\max}$) the $B_{\ell,t}$ reduces to a near-uniform mixture of exploitation and the two exploratory terms. Assuming $\beta_1 = \beta_2$ and non-drastically changing regions, as $t$ grows, the influence of $\bar{V}_{\ell,t}$ decays *faster* than that of $\overline{\mathcal{E}_{\ell,t}}$ because the latter itself shrinks with $n_\ell$. Hence, geometric exploration is front-loaded, while statistical calibration persists more throughout the optimization. When $t$ is close to $T$ the rule essentially becomes a greedy maximizer of $\bar{\mu}_{\ell,t}$, which is optimal once an $\varepsilon$-accurate maximizer has already been isolated. Thus, this composite score represents the classical trade-off: *"go where I have seen something good, go where I have not looked at all, and go where my estimate is still uncertain".*

## 3.3 SELECT: STOCHASTIC SELECTION OF PARTITIONS

Once the score $B_{\ell,t}$ equation 3 has been computed for every leaf, the algorithm must decide *where* to spend the next evaluation budget of size $b$. The Select step stochastically selects partitions by sampling from a categorical distribution over leaves. At round $t$, we draw *without replacement* a batch of $M$ distinct leaves, denoted as $\mathcal{B}_t$, from this categorical distribution where the sampling probability is: $p_{\ell,t} = B_{\ell,t} / \sum_{r=1}^{K_t} B_{r,t}$, where $\ell$ is the leaf index and $1 \leq \ell \leq K_t$. Sampling stochastically instead of selecting the top-$M$ leaves means that sub-optimal leaves are sampled infinitely often (Audibert et al., 2009), potentially helping to mitigate premature convergence especially in highly non-convex and multimodal functions. Each leaf has always a positive probability due to the small constant $\epsilon > 0$ we add to the exploitation term in Equation 1 and the min-max normalization in Equation 3. As $t$ grows, those exploratory components shrink and $B_{\ell,t}$ become increasingly peaked around the empirical best leaves, pushing $p_{\ell,t}$ toward a near-greedy regime. Moreover, a smooth annealing

---

[3]When $n_{\ell,t} < 2$, the empirical variance $\sigma_{\ell,t}^2$ is undefined or zero. To prevent a misleadingly small exploration bonus in such highly uncertain cases, $\sigma_{\ell,t}^2$ might be initialized to a small positive default value.

Figure 3: Best function value across 50 iterations on the synthetic problems. HOLLM outperforms or matches the performance of baselines.

of $\alpha_t$ in Equation 3 avoids an abrupt "switch-to-greedy" policy, which may ignore late-appearing, high-value regions if they happen to be discovered just after the switch. Finally, sampling $M$ leaves without replacement diversifies evaluations by always sampling on distinct regions.

### 3.4 Sample: LLM-Guided Candidate Generation

After the Select step has identified a batch of leaves $\mathcal{B}_t = \{\mathcal{X}_{1,t}, \ldots, \mathcal{X}_{b,t}\}$, which also contain their corresponding hyperrectangular partition boundaries, HOLLM suggests new candidate points inside each chosen partition by prompting an LLM with the following logic: *"Given the history of evaluations $\mathcal{D}_t$, propose $k$ new points that are likely to reveal high values of $f$ inside $\mathcal{X}_{i,t}$."* We construct a structured prompt (see Appendix I) containing: (i) points in $\mathcal{D}_t$ as in-context examples, (ii) the numeric partition bounds $(l_{i_s}, u_{i_s})_{s=1}^d$ for cell $\mathcal{X}_{i,t}$, and (iii) task instructions to return new proposals and their estimated function values. Feeding this prompt to the LLM yields

$$(\hat{\mathbf{x}}_i, \hat{\mathbf{f}}_i) \;=\; \text{LLM\_Generate}\big(\mathcal{D}_t, (l_{i_s}, u_{i_s})_{s=1}^d, k\big),$$

where $\hat{\mathbf{x}}_i = \{\hat{x}_{i,1}, \ldots, \hat{x}_{i,k}\} \subset \mathcal{X}_{i,t}$ and $\hat{\mathbf{f}}_i = (\hat{f}_{i,1}, \ldots, \hat{f}_{i,k}) \in \mathbb{R}^k$ are, respectively, the LLM's candidate locations and their predicted function values. We prompt the LLM to generate candidates and predict their performance with two prompts, one for generation and one for prediction. Across the $M$ selected leaves we thus obtain $k \cdot M$ suggestions. The parameter $k$ trades off the breadth of local exploration against prompt complexity and LLM inference cost. Finally, HOLLM keeps the globally best $b$ according to $\hat{f}$ and evaluates them on true function.

## 4 Empirical Evaluation

In this section, we evaluate HOLLM on a variety of search spaces and tasks. These span continuous synthetic functions, hyperparameter optimization (Feurer & Hutter, 2019; Yu & Zhu, 2020) over discrete spaces, and real world tasks.

**Baselines.** We implement HOLLM in SyneTune (Salinas et al., 2022) and compare against several different state-of-the-art algorithms found in the same framework, such as multi-fidelity methods Bayesian optimization (BO) (CQR (Salinas et al., 2023)), Gaussian Process BO with Expected Improvement (GP-EI (Snoek et al., 2012; Balandat et al., 2020)), density estimator methods (BORE (Tiao et al., 2021)), evolutionary strategies (RE (Real et al., 2019)) and random search (RS (Bergstra & Bengio, 2012)). We also ran space partitioning methods such as TurBO (Eriksson et al., 2019) and LA-MCTS (Wang et al., 2020; Yang et al., 2021) using their original implementations. We show the comparison to the last two baselines in Appendix C.1. In all benchmarks, we also compare to the global LLM-based optimizer baseline (see Algorithm 2 in the appendix) that uses the exact same prompt structure as HOLLM (we provide the prompt templates in Appendix I), with the only difference being the region boundaries. To isolate the effect of LLM-based sampling, we add an RS + KD-Tree baseline that draws candidates uniformly at random from each subregion.

**Setup.** Starting from $n_0 = 5$ initial random evaluations, we run each method 10 times for a total of $T = 50$ iterations with different random seeds and report their mean and standard error. If not stated otherwise, for HOLLM we always decay the $\alpha_t$ exploration coefficient from 1.0 to 0.01 using a cosine annealing schedule (Loshchilov & Hutter, 2017), a batch size $b = 4$, $M = 5$ selected partitions, $k = 5$ proposals per selected partition, and a fixed maximum leaf size $m_t = m_0 = \lceil d/2 \rceil$. In Appendix C.2, we provide robustness experiments when varying these hyperparameter choices in our algorithm. Unless stated otherwise, we use Gemini-2.0-Flash as our default LLM choice

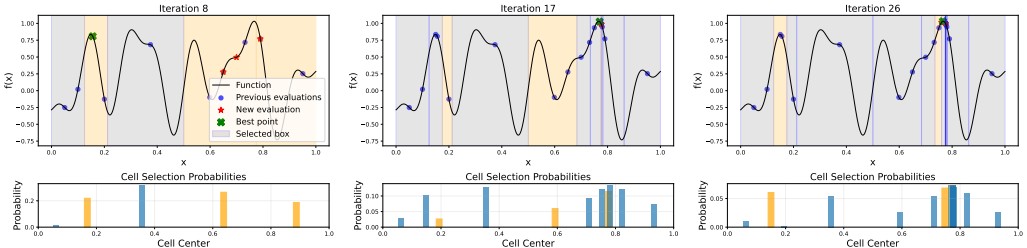

Figure 4: Illustrative example of HOLLM optimizing a 1D multimodal problem. The rectangles represent the space partitions (top figure) and are highlighted in orange whenever they are selected based on their respective probabilities (bottom figure). We used a batch size of 3. All new points (red stars) are LLM suggestions. Notice how partitions become fine-grained around the global maximum.

in `Sample` due to its fast inference speed, low cost, and large context window. Importantly, the LLM is provided with only minimal task information: the input dimensionality, variable names whenever applicable (e.g., hyperparameter names), partition boundaries, and in-context examples. No task-specific descriptions or dataset statistics are included. While prior work (Liu et al., 2024) shows that performance can improve by enriching prompts with such information, we avoid this to prevent potential contamination and reported performance on overly engineered prompts. We provide the full experimental details in Appendix B.

## 4.1 SYNTHETIC FUNCTIONS

We benchmark on six synthetic functions of varying nature and dimensionality: *Hartmann-3D* and *Hartmann-6D* (smooth but sharply multimodal), *Rosenbrock-8D* (unimodal with a narrow valley and ill-conditioning), *Rastrigin-10D* (regular multimodality with $10^{10}$ local minima) and *Lévy-10D* (plateaus, cliffs, and funnels). These functions pose challenges ranging from fine local search to broad exploration. See Table 1 in Appendix B.2 for more details on these functions. Results presented in Figure 3 show that HOLLM consistently outperforms the global LLM baseline, especially on the multimodal functions, also exhibiting less variance between runs. It also matches or surpasses all other baselines, except on Rastrigin, most likely due to the high number of local minima. In Appendix C.1, we provide additional results using Gemini-1.5-Flash, as well as a comparison to TurBO and LA-MCTS, which were implemented outside SyneTune.

**Visualizing the Optimization Process.** In Figure 4, we show a visualization of HOLLM's mechanics on a 1D multimodal function. The rectangles represent the KD-tree space partitions and they are highlighted in orange whenever they get selected. We can see that during the first iterations the partitions are larger and HOLLM is more exploratory, also confirmed by the regions' respective probabilities (bottom bar plot). Later on, as the regions become smaller, high modes are identified and by the end the score probability mass concentrates more around the global maximum. We provide similar visualizations for Lévy-1D and Rosenbrock-1D in Appendix C.

## 4.2 HYPERPARAMETER OPTIMIZATION

We assess the effectiveness of HOLLM on hyperparameter optimization by optimizing the 9D categorical space from FCNet (Klein & Hutter, 2019), where the task is to minimize the validation MSE

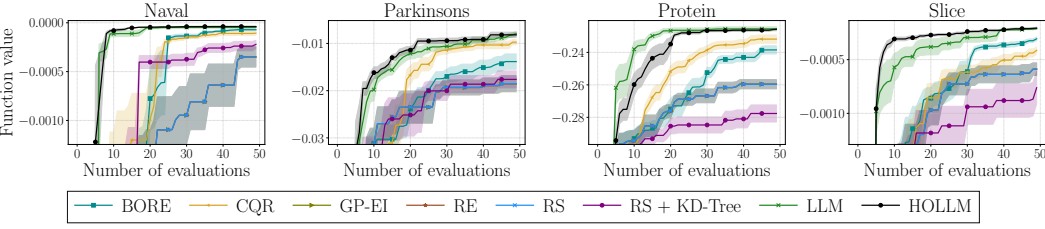

Figure 5: Hyperparameter optimization on 4 datasets from the FCNet search space. All baselines from Synetune are evaluated asynchronously using 4 workers.

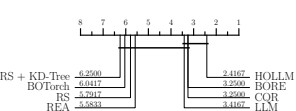

Figure 6: Results on 3 real-world tasks comparing HOLLM to baselines from SyneTune. We run each method 10 times and report the mean and standard error.

Figure 7: Critical difference diagram. Lower ranks indicate better performance.

of a fully connected network on 4 distinct datasets: PROTEIN, NAVAL, PARKINSONS and SLICE. See Appendix B.3 for more details on this search space. Results shown in Figure 5 demonstrate that both HOLLM and the global LLM optimizers outperform methods such as BORE and CQR, which typically are the off-the-shelf best choices on these benchmarks. *A single run with 50 trials of the HOLLM method has an average runtime of approximately 330 seconds, with a total cost of* $0.14 *for the Gemini-2.0-Flash API calls.* Compared to the global LLM baseline, the benefit of using space partitioning in this task is less pronounced compared to the other tasks, probably due to the discrete nature of the search space. However, HOLLM shows a clear benefit of LLM-based subregion sampling compared to the much lower performance of RS + KD-Tree.

## 4.3 REAL WORLD TASKS

To demonstrate further the practical applicability of HOLLM, we evaluate it on 3 real-world tasks from Tanabe & Ishibuchi (2020) with continuous input space: *Penicillin Production (7D)*, where the aim is to maximize penicillin yield given seven continuous variables related to fermentation; *Vehicle Safety (5D)*, where the goal is to optimize vehicle crash safety by adjusting five structural design parameters; and *Car Side Impact (7D)*, where the task is to improve side-impact performance through seven design variables. We refer the reader to Appendix B.4 for a detailed description of each of these tasks. Finally, in Figure 7 we compute the Critical Difference Diagram and show that HOLLM demonstrates the highest average rank across all tasks.

## 4.4 ABLATIONS AND ANALYSIS

**Performance with non-proprietary LLMs.** To assess the impact of the LLM model choice, we compared HOLLM against the global LLM optimizer on the Vehicle Safety task using Llama-3.1-8B (Dubey et al., 2024), Llama-3.3-70B, and Qwen3-30B-A3B (Team, 2025) as the surrogate and candidate generator LLM model. Results in Figure 8a show that HOLLM consistently found better function values and demonstrated superior robustness on the Vehicle Safety benchmark. While the standard optimizer failed by stalling immediately when paired with the Llama models, HOLLM successfully used these same models to achieve performance approaching that of Gemini-2.0-Flash. In Appendix H we present the results on the other real-world benchmarks.

**Ablation Study: Scoring, Selection, and Sampling.** To justify the HOLLM scoring function, we conduct an ablation study on *Vehicle Safety* (Gemini-2.0-Flash) isolating key components: (i) HOLLM (UCB1), replacing UCB-V with UCB1 (Auer et al., 2002); (ii) HOLLM (Exploitation Only), removing the exploration bonus; (iii) HOLLM (Exploration Only), removing the exploitation term; and (iv) HOLLM (Uniform), selecting regions uniformly. Results (Figure 8b) highlight the importance of variance-aware exploration: UCB1 over-explores stable but suboptimal regions, while UCB-V adapts to empirical variance, enabling faster identification of high-potential subregions; replacing UCB-V with UCB1 reduces convergence speed and final performance. The full scoring function outperforms both Exploitation-only and Exploration-only variants, confirming the need for a balanced tradeoff, while the Uniform baseline performs worst, indicating that HOLLM's gains stem from intelligent budget allocation via the scoring function rather than from partitioning or LLM capability alone. Additional results on *NAS-Bench-201* (Dong & Yang, 2020) (Gemini-1.5-Flash) are provided in Appendix E.

**Robustness to Observation Noise.** Another key advantage of integrating a variance-sensitive scoring function (UCB-V) with localized partitioning is the ability to handle noisy objective functions. To evaluate this robustness aspect, we ran HOLLM and the baselines on the *Vehicle Safety* benchmark

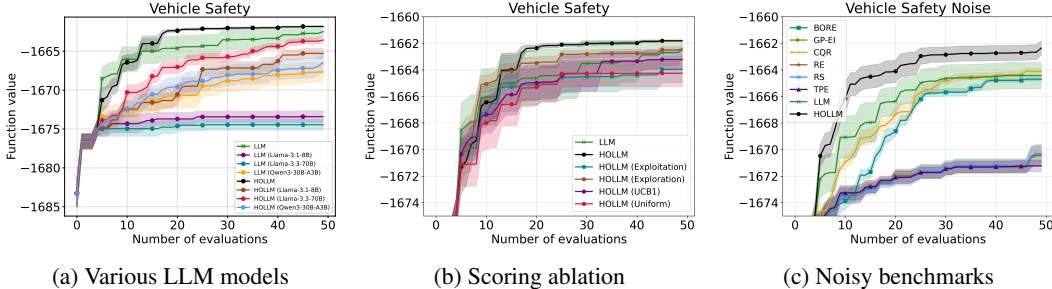

(a) Various LLM models          (b) Scoring ablation          (c) Noisy benchmarks

Figure 8: **Analysis & Ablations on Vehicle Safety.** (a) Performance when using various LLM models. (b) Ablation of the scoring function: we compare the HOLLM scoring function (black) against variants replacing UCB-V with UCB1, using only exploration or exploitation terms, and selecting regions uniformly at random. (c) Performance on noisy benchmarks: HOLLM demonstrates superior robustness and sample efficiency compared to the global LLM baseline and traditional methods when the function is corrupted by $\mathcal{N}(0, \sigma^2)$ noise ($\sigma$ is 1% of the global function range).

under observation noise. The noise was modeled as zero-mean homoscedastic Gaussian noise ($\mathcal{N}(0, \sigma^2)$), where the standard deviation ($\sigma$) was fixed at 1% of the global objective function range. As illustrated in Figure 8c, HOLLM exhibits a clear and significant performance advantage over all baselines. While the global LLM baseline and standard methods like TPE (Bergstra et al., 2011) and Random Search (Bergstra & Bengio, 2012) struggle to filter the noise and plateau prematurely, HOLLM consistently reaches a higher objective value. By restricting the context locally, HOLLM obtains more reliable mean estimates even under heavy observation noise, ensuring that exploitation remains focused and robust.

**Runtime Complexity, Surrogate Performance & Sample Diversity.** In Appendix G, we discuss the runtime complexity and sample efficiency of HOLLM. Moreover, in Appendix F and Appendix D, we study the surrogate performance of the LLM in comparison to Gaussian Processes (Rasmussen & Williams, 2006) and the recent TabPFN-v2 (Hollmann et al., 2025), as well as the sample diversity of HOLLM vs. the global LLM.

## 5    CONCLUSION, LIMITATIONS AND SOCIETAL IMPACT

We propose HOLLM, a novel LLM-based global optimization method for expensive blackbox functions, that combines adaptive KD-tree partitioning, a bandit-inspired score function, and LLM capabilities for generating new candidate points on locally promising regions. HOLLM excels especially on multimodal functions with many local optima that pose a risk for premature convergence, hyperparameter optimization and real-world tasks, consistently outperforming LLMs with a global sampling policy and other non-LLM state-of-the-art methods.

**Limitations.** While HOLLM combines nonparametric bandit methods with LLM-based sampling and shows strong empirical performance, it has several limitations. Firstly, its effectiveness depends heavily on the quality of LLM-generated proposals; biased or miscalibrated models can misguide the search or waste evaluations. Secondly, the inference and monetary cost of LLMs, especially proprietary ones, can limit scalability in high-dimensional settings. Third, although default parameter values perform well in our experiments, real-world deployment may require tuning them to avoid premature convergence or excessive exploration. Finally, our approach currently lacks formal theoretical guarantees, especially regarding regret bounds, which we leave for the future.

**Impact.** The use of LLMs in global optimization has several societal implications. HOLLM has the potential to accelerate progress in areas such as drug discovery, materials design, and energy systems by reducing experimental costs and enabling personalized solutions. On the other hand, reliance on LLMs trained on biased data risks perpetuating social injustices when guiding sensitive decisions (e.g., hiring). Additionally, repeated LLM queries incur considerable energy costs, and the opacity of LLM-driven decisions may limit transparency and reproducibility. Therefore, responsible deployment requires bias assessment, usage controls, and transparency in both computational and ethical impacts.

## ACKNOWLEDGMENTS

We acknowledge funding by the European Union (via ERC Consolidator Grant DeepLearning 2.0, grant no. 101045765). Views and opinions expressed are however those of the author(s) only and do not necessarily reflect those of the European Union or the European Research Council. Neither the European Union nor the granting authority can be held responsible for them. Frank Hutter acknowledges the financial support of the Hector Foundation. We also thank Google Cloud for their free trial program, that enabled us to use the Google Gemini models throughout this project.

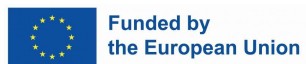

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

## A ALGORITHM PSEUDOCODES

In this section, we present detailed pseudocode for the HOLLM algorithm in Algorithm 3, which complements Algorithm 1. In the algorithm, we omit the subscript $t$ for easier readability. Additionally, Algorithm 2 describes the LLM-based global optimization baseline method used throughout our experiments. To ensure fair comparison, we configure the baseline to propose $k \cdot b$ points per iteration (line 2 of Algorithm 2), matching the total number of proposals generated by HOLLM across all subregions, i.e., $k$ proposals in each of the $M$ subregions.

---

**Algorithm 2:** GLOBAL-LLM baseline

    **Data:** Initialize $\mathcal{D}$ with $n_0$ points, budget $T$, batch size $b$, proposals $k \cdot M$

1. **for** $t = n_0, \ldots, T - 1$ **do**
2.     Propose $k \cdot M$ points with LLM_GENERATE$(\mathcal{X}, \mathcal{D}, k \cdot M)$
3.     Evaluate the top $b$ and add to $\mathcal{D}$
4. **return** $best \; (x, y) \in \mathcal{D}$

---

## B DETAILS ON TASKS, BASELINES AND EXPERIMENTAL SETUP

### B.1 BASELINES

We compare HOLLM to the following baselines:

- **Random Search (RS) Bergstra & Bengio (2012)** serves as a simple baseline that uniformly samples configurations from the search space without any learning or adaptation.

- **Regularized Evolution (RE) Real et al. (2019)** is an evolutionary algorithm that maintains a population of candidate solutions and evolves them through mutation operations. The method regularizes the population by removing the oldest individuals, preventing stagnation and maintaining diversity.

- **Conformalized Quantile Regression (CQR) Salinas et al. (2023)** uses gradient boosted trees to predict performance quantiles and provides prediction intervals with statistical guarantees through conformal prediction techniques.

- **Bayesian Optimization by Density-Ratio Estimation (BORE) Tiao et al. (2021)** reformulates Bayesian optimization as a binary classification problem. It trains a probabilistic classifier to distinguish between high-performing and low-performing configurations, then uses the predicted class probabilities to construct an acquisition function equivalent to expected improvement.

- **Gaussian Process BO with Expected Improvement (GP-EI) Snoek et al. (2012); Balandat et al. (2020)** employs a Gaussian Process as a surrogate model to capture the objective function's behavior and uncertainty. It uses the Expected Improvement acquisition function, implemented via BoTorch, to balance exploration and exploitation when selecting new evaluation points.

### B.2 SYNTHETIC BENCHMARKS

In Section 4, we initially evaluated HOLLM and the baselines on 6 synthetic deterministic function with varying dimensionality and nature. In Table 1, we provide details on each of them.

### B.3 HYPERPARAMETER OPTIMIZATION BENCHMARKS

For our hyperparameter optimization experiments, we evaluate on four tasks from the FCNet benchmark Klein & Hutter (2019): PROTEIN, NAVAL, PARKINSONS, and SLICE. The FCNet benchmark provides a tabulated hyperparameter optimization setting where fully connected neural networks are trained on each dataset with different hyperparameter configurations. The search space consists of 9 categorical hyperparameters (network architecture and training parameters), yielding 62,208 possible configurations with pre-computed validation accuracies. To enable KD-tree partitioning on the categorical search space, we apply ordinal encoding to convert categorical variables into numerical

---

**Algorithm 3:** HIERARCHICAL OPTIMIZATION WITH LLMS (HOLLM) – DETAILED

---

**Data:** Initial data $\mathcal{D} = \{(x_i, f(x_i))\}_{i=1}^{n_0}$, batch size $b$, regions to sample from $M$, proposal count per leaf $k$, dimension $d$, initial leaf size $m_0$, adaptive growth rate $\lambda$, total evaluations $T$, exploration weights $\beta_1, \beta_2$, annealing $\alpha_{min}, \alpha_{max}$

1.   $t \leftarrow n_0$                                          `// global evaluation counter`

2.   **while** $t < T$ **do**

3.       (optional) $m_{\text{leaf}} \leftarrow m_0 + \lceil \lambda \log(1 + t) \rceil$            `// adaptive leaf size`

4.       Build KD-tree on $\mathcal{D}$ with leafsize $m_{\text{leaf}}$, obtaining $K_t$ leaves $\{\mathcal{X}_\ell\}_{\ell=1}^{K_t}$

5.       $\alpha \leftarrow \alpha_{min} + \frac{1}{2}(\alpha_{max} - \alpha_{min})(1 + \cos(\pi t/T))$      `// cosine annealing` schedule

6.       $f_{\min} \leftarrow \min_{i=1}^{t} f(x_i)$

7.       $Y_+ \leftarrow \{f(x_i) - f_{\min} + \epsilon\}_{i=1}^{t}$           `// positive transformed values`

8.       **for** $l = 1$ **to** $K_t$ **do**

9.           $n_\ell \leftarrow |\mathcal{X}_\ell|$                 `// number of points in this leaf`

10.           $(l_\ell, u_\ell) \leftarrow$ bounds of cell $\mathcal{X}_\ell$

11.           $V_\ell \leftarrow \prod_{j=1}^{d}(u_{\ell j} - l_{\ell j})^{1/d}$           `// normalized volume`

12.           **if** $n_\ell > 0$ **then**

13.               $\mu_\ell \leftarrow \max_{i \in I_\ell} Y_+[i]$         `// best value in cell`

14.               **if** $n_\ell > 1$ **then**

15.                  $\sigma_\ell^2 \leftarrow \frac{1}{n_\ell - 1} \sum_{i \in I_\ell} (Y_+[i] - \bar{Y}_\ell)^2$      `// variance in cell`

16.               **else**

17.                  $\sigma_\ell^2 \leftarrow 0.01$    `// default variance for single-point cells`

18.               $\log\_term \leftarrow \max(0, \log(t/(K_t \cdot n_\ell)))$

19.               $\mathcal{E}_\ell \leftarrow (\sqrt{2\sigma_\ell^2 \cdot \log\_term/n_\ell} + \log\_term/n_\ell)$

20.           **else**

21.               $\mu_\ell, V_\ell, \mathcal{E}_\ell \leftarrow 0, 1, 1$       `// high exploration for empty cells`

22.           $\bar{\mu}, \bar{V}_\ell, \bar{\mathcal{E}}_\ell \leftarrow$ min-max normalize $\mu_\ell, V_\ell, \mathcal{E}_\ell$ across all cells

23.           $B_\ell \leftarrow \bar{\mu} + \alpha \cdot (\beta_1 \cdot \bar{V}_\ell + \beta_2 \cdot \bar{\mathcal{E}}_\ell)$         `// composite score`

24.       $p_\ell = B_\ell / \sum_{r=1}^{K_t} B_r$           `// normalize score across cells`

25.       Sample $M$ cells $\{\mathcal{X}_{i_j}\}_{j=1}^{M} \sim \text{Categorical}(\{p_\ell\})$

26.       $\hat{X} \leftarrow \emptyset, \ \hat{F} \leftarrow \emptyset$

27.       **for** $j = 1$ **to** $M$ **do**

28.           $(\hat{\mathbf{x}}_j, \hat{\mathbf{f}}_j) \leftarrow \text{LLM\_GENERATE}(\mathcal{D}, (l_{i_j}, u_{i_j}), k)$

29.           Append $\hat{\mathbf{x}}_j$ to $\hat{X}$; append $\hat{\mathbf{f}}_j$ to $\hat{F}$

30.       $\pi \leftarrow \text{argsort}(\hat{F})$        `// indices of sorted values (descending)`

31.       $X^{\text{new}} \leftarrow$ top $b$ points from $\hat{X}$ using indices $\pi$

32.       **for** *each* $x \in X^{new}$ **do**

33.           Evaluate $y = f(x)$

34.           $\mathcal{D} \leftarrow \mathcal{D} \cup \{(x, y)\}$

35.           $t \leftarrow t + 1$

36.   **return** *best point* $(x^*, f(x^*))$ *where* $x^* = \arg\max_{x \in \mathcal{D}} f(x)$

---

Table 1: List of synthetic optimization functions and their main characteristics.

| Function (dim.) | Landscape & key traits | Global boundary | Global optimum |
|---|---|---|---|
| Hartmann 3D | Smooth, strongly multimodal surface generated by four weighted Gaussians inside the unit cube; narrow, steep basins punish local search. | $(x_1, x_2, x_3) \in [0, 1]^3$ | $f_{\min} \approx -3.86278$ |
| Hartmann 6D | Six Gaussians in $[0, 1]^6$ create an even denser constellation of deceptive wells; still smooth but mildly ill-conditioned, and the search space grows exponentially. | $(x_1, x_2, \ldots, x_6) \in [0, 1]^6$ | $f_{\min} \approx -3.32237$ |
| Rosenbrock 8D | Classic curved "banana" valley stretched to eight variables; unimodal yet highly ill-conditioned, requiring precise valley-tracking; non-separable. | $(x_1, x_2, \ldots, x_8) \in [-2.048, 2.048]^8$ | $f_{\min} = 0$ |
| Rastrigin 10D | Quadratic core overlaid with cosine ripples forms a perfectly regular grid of $10^{10}$ local minima; separable but brutally multimodal, exposing algorithms prone to premature convergence. | $(x_1, x_2, \ldots, x_{10}) \in [-5.12, 5.12]^{10}$ | $f_{\min} = 0$ |
| Lévy 10D | Sine perturbations on a quadratic backbone yield wide plateaus, sudden cliffs, and deep funnels—rugged and non-separable, stressing step-size control. | $(x_1, x_2, \ldots, x_{10}) \in [-10, 10]^{10}$ | $f_{\min} = 0$ |
| Ackley 20D | Exponential of radius plus averaged cosines: vast flat outer region, encircling ridge, and a single sharp basin at the origin; tests exploration versus exploitation in very high dimension. | $(x_1, x_2, \ldots, x_{20}) \in [-32.768, 32.768]^{20}$ | $f_{\min} = 0$ |

split indices. Below we describe the four regression datasets used as the underlying machine learning tasks:

- PROTEIN is a regression dataset containing physicochemical properties of protein tertiary structures. The task involves predicting protein properties from 9 molecular descriptors across 45,730 protein samples.

- PARKINSONS contains biomedical voice measurements from 42 individuals with early-stage Parkinson's disease participating in a six-month telemonitoring trial. The regression target is the progression of Parkinson's symptoms, with 5,875 samples and 19 acoustic features.

- NAVAL consists of simulated sensor data from a naval frigate's propulsion system, including gas turbine, propeller, gearbox, and control systems. The regression task predicts component degradation levels using 11,934 samples with 16 operational features.

- `SLICE` involves predicting the relative axial location of CT scan slices within the human body. The dataset contains 384 features extracted from 53,500 CT images, describing bone structures, air inclusions, and anatomical positioning.

## B.4   REAL-WORLD OPTIMIZATION TASKS

To evaluate the practical applicability of HOLLM, we consider three real-world global optimization problems from Tanabe and Ishibuchi Tanabe & Ishibuchi (2020). These tasks are commonly used benchmarks that originate from engineering and bioprocess applications, each formulated as a continuous blackbox optimization problem with simulators of real systems. Below, we provide detailed descriptions of each task and in Table 3 the input parameter names for each task and their respective value ranges.

**Penicillin Production.** This task models the optimization of a penicillin fermentation process. The objective is to maximize the penicillin yield, by searching in a 7D continuous design space, consisting of culture volume, biomass concentration, temperature, glucose concentration, substrate feed rate, substrate feed concentration, and hydrogen ion ($H^+$) concentration. This problem formulation has been widely used as a benchmark in global optimization due to its nonlinear dynamics and practical relevance in bioprocess engineering.

Table 2: Search space of the FCNet benchmark, listing each hyperparameter and its possible categorical choices.

| Hyperparameter | Categorical Configuration Space |
| --- | --- |
| Initial LR | {0.0005, 0.001, 0.005, 0.01, 0.05, 0.1} |
| Batch Size | {8, 16, 32, 64} |
| LR Schedule | {cosine, fix} |
| Activation (Layer 1) | {relu, tanh} |
| Activation (Layer 2) | {relu, tanh} |
| Layer 1 Size | {16, 32, 64, 128, 256, 512} |
| Layer 2 Size | {16, 32, 64, 128, 256, 512} |
| Dropout (Layer 1) | {0.0, 0.3, 0.6} |
| Dropout (Layer 2) | {0.0, 0.3, 0.6} |

**Vehicle Safety.** This task models the design of a vehicle's frontal structure to improve crash safety. The primary objective is to maximize crashworthiness, evaluated in terms of structural performance under frontal impact. The problem is formulated as a 5D continuous optimization problem, where the decision variables correspond to the thicknesses of different structural components in the front of the vehicle. These parameters directly influence crash responses such as weight, acceleration, and deformation of the passenger compartment, making it a well-established surrogate problem for safety-critical vehicle design.

**Car Side Impact.** This task models vehicle performance under a lateral collision. The primary objective is to improve side-impact safety through careful adjustment

Table 3: List of real-world optimization tasks and their main characteristics.

| Task | Input variable | Range |
| --- | --- | --- |
| Penicillin Production (7D) | Culture volume | (60.0, 120.0) |
| | Biomass concentration | (0.05, 18.0) |
| | Temperature | (293.0, 303.0) |
| | Glucose concentration | (0.05, 18.0) |
| | Substrate feed rate | (0.01, 0.5) |
| | Substrate feed concentration | (500.0, 700.0) |
| | $H^+$ concentration | (5.0, 6.5) |
| Vehicle Crashworthiness (5D) | Thickness of member 1 | (1.0, 3.0) |
| | Thickness of member 2 | (1.0, 3.0) |
| | Thickness of member 3 | (1.0, 3.0) |
| | Thickness of member 4 | (1.0, 3.0) |
| | Thickness of member 5 | (1.0, 3.0) |
| Car Side Impact (7D) | B-pillar thickness | (0.5, 1.5) |
| | Door beam thickness | (0.45, 1.35) |
| | Floor side thickness | (0.5, 1.5) |
| | Cross-members thickness | (0.5, 1.5) |
| | Door thickness | (0.875, 2.625) |
| | Roof rail thickness (front) | (0.4, 1.2) |
| | Roof rail thickness (rear) | (0.4, 1.2) |

of structural design parameters. It is a 7D continuous optimization problem, where the decision variables correspond to aspects of the car body that affect passenger safety during side impacts.

# C ADDITIONAL EXPERIMENTS

In this section, we provide additional experiments and ablations, complementing the ones conducted throughout Section 4 of the main paper.

## C.1 RESULTS USING GEMINI-1.5-FLASH

In this section, we present some preliminary results we obtained by running HOLLM and the global LLM optimizer with Gemini-1.5-Flash on 6 synthetic benchmarks, the FCNet hyperparameter optimization tasks, and neural architecture search on NAS-Bench-201 (Dong & Yang, 2020).

### C.1.1 SYNTHETIC BENCHMARKS

Figure 9 shows the results when running HOLLM (Gemini-1.5-Flash) and the other baselines on the same synthetic benchmarks as in Section 4. We additionally ran on *Ackley-20D*, which is characterized by flat regions and a single sharp global minimum at the origin. We ran HOLLM and all other baselines 3 times with different random seeds for 100 iterations. Similarly as with Gemini-2.0-Flash, HOLLM consistently outperforms or matches the baselines, even on Rastrigin. Most notably, on Ackley-20D with input range $[-32.768, 32.768]^{20}$, HOLLM locates the global maximum in just 50 iterations, while baselines struggle to improve beyond random search. However, more repetitions are necessary to obtain more conclusive results from this experiment.

Additionally, we evaluated space partitioning methods such as TurBO (Eriksson et al., 2019) and LA-MCTS (Wang et al., 2020; Yang et al., 2021), as well as other popular blackbox optimizers such as TPE (Bergstra et al., 2011), SMACv3 (Lindauer et al., 2022), CMA-ES (Hansen & Auger, 2011) and HEBO (Cowen-Rivers et al., 2022). For TurBO, LA-MCTS, CMA-ES and HEBO, we use their official implementations, whilst SMACv3 and TPE are in SyneTune. Table 4 shows the mean and standard error of 5 runs after 100 iterations on 3 synthetic benchmarks. As it can be seen, HOLLM is still superior compared to all baselines, including the space partitioning ones. Moreover, even when running LA-MCTS for 1000 function evaluations, it still is not able to find the global minimum in Ackley 20D, while HOLLM does that in less than 50 evaluations.

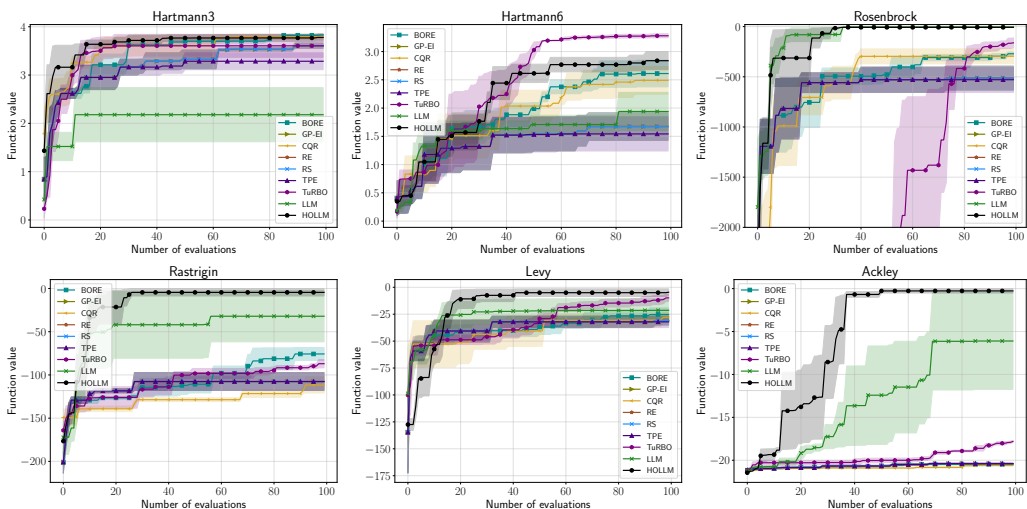

Figure 9: Best function value across 100 iterations on the synthetic problems. HOLLM outperforms or matches the performance of baselines, especially on higher dimensional problems (e.g., Ackley). For this experiment we used Gemini-1.5-Flash and the results show the mean and standard error of 3 runs.

### C.1.2 HYPERPARAMETER OPTIMIZATION

In Figure 10, we show the results HOLLM compared to the same baselines as in the experiments in Section 4 of the main paper, however, this time using Gemini-1.5-Flash as the LLM model inside

Table 4: Comparison of the performance of HOLLM and the global LLM optimizer using Gemini-1.5-Flash to other non-LLM-based global optimization methods. We ran each method for 100 trials (or 1000 when specified so) and report the mean and standard error of 5 repetition.

| Method | Rosenbrock 8D | Rastrigin 10D | Ackley 20D |
|---|---|---|---|
| CMA-ES | -970.56 ± 689.68 | -125.85 ± 9.18 | -20.96 ± 0.15 |
| CMA-ES (1000 iters) | -117.21 ± 35.05 | -67.12 ± 6.52 | -19.24 ± 1.16 |
| CQR | -295.79 ± 17.04 | -111.68 ± 2.14 | -20.59 ± 0.03 |
| BORE | -270.70 ± 4.03 | -75.73 ± 1.73 | -20.49 ± 0.01 |
| GP-EI | -512.86 ± 27.64 | -107.63 ± 2.39 | -20.39 ± 0.01 |
| RE | -512.86 ± 27.64 | -107.63 ± 2.39 | -20.39 ± 0.01 |
| RS | -512.86 ± 27.64 | -107.63 ± 2.39 | -20.39 ± 0.01 |
| RS (1000 iters) | -194.78 ± 67.66 | -91.65 ± 6.21 | -20.05 ± 0.17 |
| TPE | -529.35 ± 29.05 | -107.63 ± 2.39 | -20.39 ± 0.01 |
| SMAC | -298.60 ± 7.15 | -104.40 ± 1.19 | -20.09 ± 0.03 |
| HEBO | -7.22 ± 3.07 | -49.63 ± 8.54 | -18.07 ± 2.29 |
| TuRBO | -161.61 ± 11.23 | -87.00 ± 1.11 | -17.81 ± 0.03 |
| LA-MCTS (RS) | -212.10 ± 69.59 | -124.71 ± 45.59 | -18.91 ± 0.45 |
| LA-MCTS (RS) (1000 iters) | -22.19 ± 6.42 | -57.05 ± 6.02 | -19.11 ± 0.39 |
| LA-MCTS (BO) | -236.23 ± 79.80 | -102.53 ± 31.96 | -17.75 ± 0.85 |
| LA-MCTS (BO) (1000 iters) | -26.60 ± 10.71 | -43.59 ± 6.00 | -13.16 ± 0.65 |
| LA-MCTS (TuRBO) | -98.36 ± 73.54 | -50.67 ± 12.46 | -16.33 ± 1.89 |
| LA-MCTS (TuRBO) (1000 iters) | -27.33 ± 24.40 | -14.16 ± 8.65 | -3.51 ± 1.01 |
| LA-MCTS (CMA-ES) | -30.56 ± 27.37 | -67.28 ± 10.43 | -9.04 ± 0.42 |
| LA-MCTS (CMA-ES) (1000 iters) | -6.31 ± 1.16 | -25.10 ± 16.12 | -1.71 ± 1.06 |
| **LLM** | **-4.66 ± 0.50** | -32.08 ± 6.84 | -6.10 ± 1.30 |
| **HOLLM (ours)** | -7.03 ± 0.01 | **-0.17 ± 0.04** | **-0.00 ± 0.00** |

HOLLM and the global LLM baseline. We ran each optimizer 3 times with different seeds and report the mean and standard error after 100 iterations.

### C.1.3 NEURAL ARCHITECTURE SEARCH

Neural Architecture Search (NAS) (White et al., 2023), like hyperparameter optimization, aims to identify the best-performing neural network architecture for a given dataset by maximizing validation accuracy. We use the NAS-Bench-201 benchmark (Dong & Yang, 2020), which provides precomputed validation accuracies for all architectures on CIFAR-10, CIFAR-100, and Downsampled ImageNet $16 \times 16$ (Chrabaszcz et al., 2017). This tabulated format enables efficient benchmarking by eliminating the computational overhead of training each architecture from scratch. The search space is 6D, with each dimension representing a discrete choice among 5 possible layer operations: `avg_pool_3x3` (average pooling), `nor_conv_3x3` (normal 3×3 convolution), `skip_connect` (identity connection), `nor_conv_1x1` (normal 1×1 convolution), and `none` (no operation).

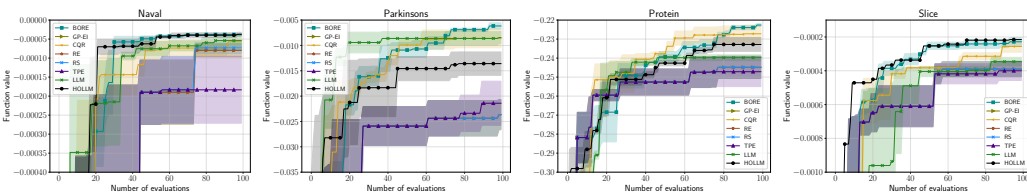

Figure 10: Hyperparameter optimization on 4 datasets from the FCNet search space. All baselines from Synetune are evaluated asynchronously using 4 workers. For this experiment we used Gemini-1.5-Flash and the results show the mean and standard error of 3 runs.

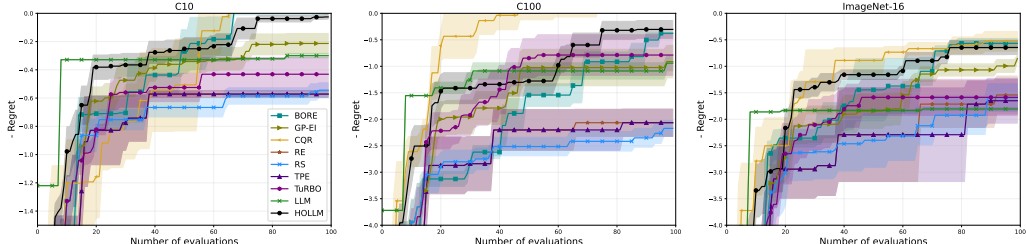

Figure 11: Results on the NAS-Bench-201 6 dimensional discrete function. We plot the negative regret vs. the number of iterations. We run each method 6 times and report the mean and standard error.

We ran experiments using Gemini-1.5-Flash. We also use a continuous representation $[0,1]^6$ of the input space and discretize it to evaluate the true function. As seen in Figure 11, HOLLM always outperforms the LLM baseline that samples globally and is on par with BORE and CQR. The global LLM seems to get stuck in local minima, therefore leading to stagnating performance from early on.

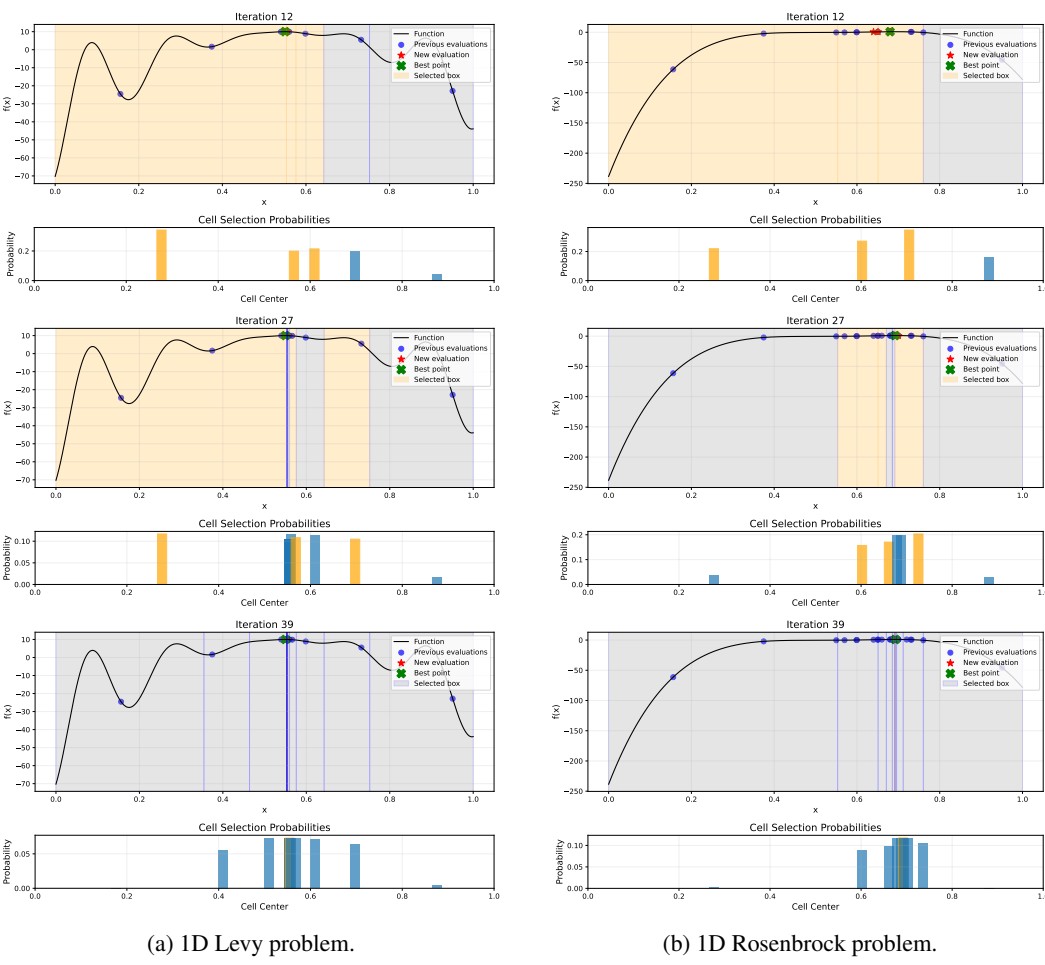

(a) 1D Levy problem.

(b) 1D Rosenbrock problem.

Figure 12: Illustrative examples of optimizing 1D functions. The rectangles represent the input space partitions (top row). They are highlighted in orange whenever selected based on their respective probabilities (bottom row). Both experiments used 5 initial points and a batch size of 4. All new points (red stars) are suggestions from the LLM.

## C.2 ABLATIONS AND FURTHER ANALYSIS

To assess the robustness of our method and understand the influence of key hyperparameters on performance, we conducted a comprehensive ablation study. We employ the 10D Levy test function and examine 3 hyperparameters that directly impact the exploration-exploitation balance and efficacy of our approach: (i) maximum leaf capacity $m_{\text{leaf}} = m_0 + \lceil \lambda \log(1 + t) \rceil$ ($\lambda = 0$), which controls the granularity of space partitioning; (ii) candidate sampling rate $k$ (proposals generated per selected region), which determines the diversity of proposals within each selected region; and (iii) region selection parameter $M$ (partitions selected per iteration), which governs the number of promising subregions explored simultaneously per iteration. The default hyperparameter configuration also used throughout the experiments in the main paper is: exploration parameter bounds $\alpha_{\max} = 1.0$ and $\alpha_{\min} = 0.01$, initial random sampling phase of $n_0 = 5$ evaluations, batch size $b = 4$ (points evaluated per iteration), $k = 5$, $M = 5$, and maximum leaf capacity $m_0 = d/2$, where $d$ denotes problem dimensionality. We run each setting with 5 independent random seeds and report the mean performance $\pm$ standard error in Figure 13.

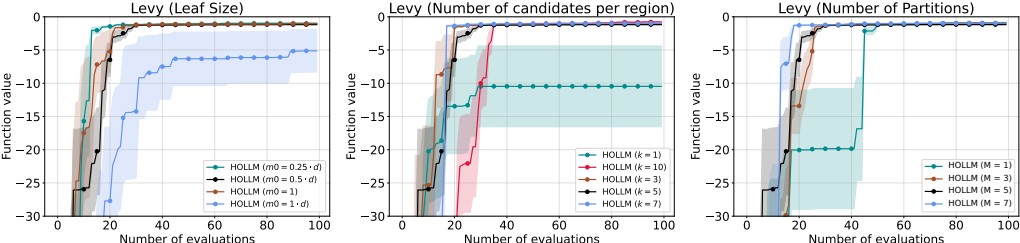

Figure 13: Impact of key hyperparameters on optimization performance for the 10D Levy function. **Left:** Ablation on leaf size ($m_0$) demonstrates that coarse partitioning ($m_0 = 1d$) yields the poorest performance, while finer partitioning enables superior exploitation of promising regions through more granular space decomposition. **Middle:** Varying the number of candidates per selected region ($k$) reveals optimal performance at intermediate values, where undersampling ($k = 1$) significantly degrades performance by limiting exploitation of high-potential regions, and oversampling ($k = 10$) slows convergence due to inefficient allocation of evaluation budget. **Right:** The number of partitions selected per trial ($M$) governs exploration breadth, where single-region focus ($M = 1$) impedes convergence through insufficient exploration, while moderate values ($M \in \{3, 5, 7\}$) accelerate optimization by enabling simultaneous exploration of multiple promising regions.

**Impact of Leaf Size** ($m_0$). The leaf size parameter $m_0$ defines the maximum number of data points within a single leaf of the partitioning tree, directly controlling the granularity of search space decomposition. Our analysis across different values of $m_0$ as a factor of problem dimensionality $d$ reveals a clear trade-off between partition resolution and statistical reliability (Figure 13, *left*). Coarse partitioning with $m_0 = d$ yields suboptimal performance due to overly broad regions that group diverse areas of the search space, diminishing the method's ability to precisely isolate promising subregions. Conversely, extreme fine partitioning with $m_0 = 1$ also degrades performance because singleton regions provide insufficient statistical information and the variance component becomes a small constant across all regions, eliminating valuable uncertainty estimates necessary to guide exploration. We observe the best performance at $m_0 = d/4$, which strikes an effective balance by enabling detailed space partitioning while maintaining sufficient data density within each region to compute meaningful variance estimates for the exploration term.

**Impact of Number of Candidates per Region** ($k$). We investigated the effect of varying the number of candidate points $k$ sampled from each selected region, testing values $k \in \{1, 3, 5, 7, 10\}$. Results shown in Figure 13, *middle*) reveal a clear trade-off between under- and over-sampling within regions. Setting $k = 1$ leads to significant performance degradation as the method fails to adequately exploit promising regions by drawing only a single sample per region. Conversely, $k = 10$ results in worse performance during initial iterations compared to intermediate $k$ values, which we attribute to increased risk of oversampling sub-optimal regions in the beginning. While the method can recover from this scenario as oversampled sub-optimal regions receive lower scores in subsequent iterations, the initial performance penalty demonstrates that excessive sampling can be counterproductive. These findings showcase the importance of balanced exploitation within selected

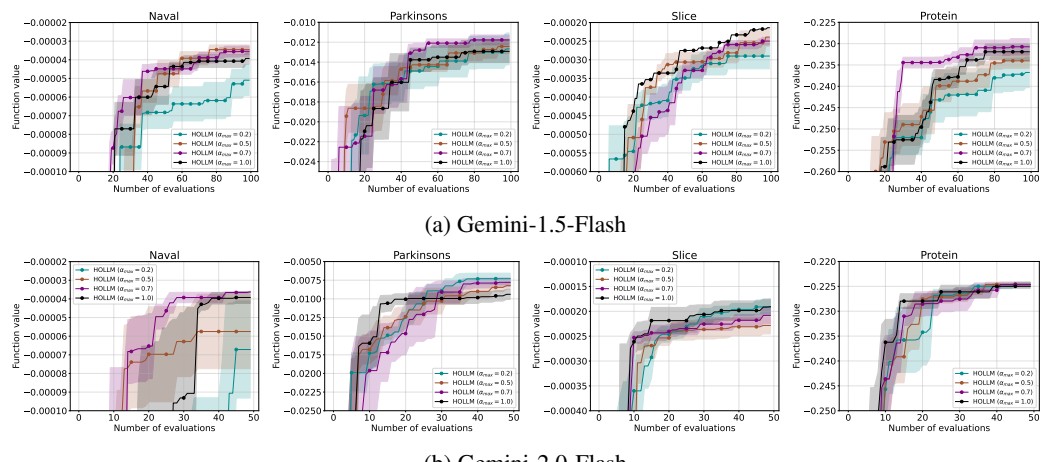

(a) Gemini-1.5-Flash

(b) Gemini-2.0-Flash

Figure 14: Performance comparison of exploration parameter settings ($\alpha_{\max} \in \{0.2, 0.5, 0.7, 1.0\}$) across four FCNet benchmark tasks. Each curve shows the mean objective function value over 5 runs, with shaded regions denoting standard error. Results illustrate that the optimal choice of $\alpha_{\max}$ is task dependent, reflecting different landscape characteristics and exploration needs.

regions: sufficient sampling to capitalize on promising areas without overcommitting computational budget to potentially sub-optimal regions.

**Impact of Number of Partitions Selected per Trial ($M$).** We examined how the number of partitions $M \in \{1, 3, 5, 7\}$ selected per trial affects optimization performance. As seen in Figure 13, *right*, setting $M = 1$ notably hinders performance, particularly during initial iterations, as HOLLM severely restricts exploration breadth by focusing all sampling efforts on a single region per iteration. When the initially chosen region lacks global promise, progress becomes slow, though the method can eventually exploit good regions once identified, leading to convergence that is typically slower than broader exploration strategies. Conversely, increasing $M$ to moderate or high values (3, 5, or 7) generally improves initial exploration by enabling simultaneous consideration of multiple diverse regions. This expansion of the candidate selection pool allows the algorithm to benefit from a larger, more diverse set of proposals per trial, improving performance up to a saturation point. The results demonstrate that balanced multi-region exploration through appropriate $M$ values provides superior performance compared to overly focused single-region strategies, highlighting the importance of maintaining exploration breadth while preserving the ability to exploit promising areas effectively.

### C.2.1 IMPACT OF EXPLORATION PARAMETER $\alpha_{max}$

We evaluated the effect of different exploration settings on the FCNet benchmarks across four tasks: PROTEIN, NAVAL, PARKINSONS, and SLICE. Results in Figure 14 show that the impact of the exploration parameter $\alpha_{\max}$ exhibits task-dependent variation, with optimal settings determined by the underlying problem structure. Higher values of $\alpha_{\max}$ bias the search toward less-explored regions, proving beneficial for highly multimodal or non-convex landscapes where diverse exploration is crucial for escaping local optima. Conversely, lower $\alpha_{\max}$ values reduce exploration of new regions and concentrate search efforts on exploitation, which may be more appropriate for smoother or convex solution spaces where intensive local search around promising areas yields better returns. These findings suggest that prior knowledge of the task landscape characteristics, such as modality, convexity, and noise structure, can effectively guide the selection of $\alpha_{\max}$ to match the exploration-exploitation balance to the problem's inherent difficulty and structure.

### C.2.2 EFFECT OF HYPERPARAMETERS ON COMPUTATIONAL COST

The computational complexity of our method scales directly with the total number of candidate points generated per iteration, calculated as $k \cdot M$. When employing computationally expensive models such as large language models accessed via API calls or hosted locally, increases in either $M$ (which amplifies the number of inference calls per iteration) or $k$ (which extends the number of output tokens

generated per call) result in proportionally higher inference times and associated costs per optimization step. This creates a fundamental trade-off between optimization performance and computational efficiency that practitioners must carefully consider based on their specific resource constraints and performance requirements. Our default configuration of $M = 5$ and $k = 5$, yielding 25 candidate evaluations per iteration, represents a calibrated compromise that balances exploration capability with computational practicality across the diverse benchmarks in our experimental evaluation.

## D  SAMPLE DIVERSITY ANALYSIS

To better understand the performance gap between HOLLM and the global LLM baseline, we analyze the *sampling diversity* of the new proposals generated during optimization. Our goal is to investigate and quantify how strongly each method is biased towards previously observed points versus exploring new regions of the search space. To this end, we introduce the *ICL Divergence* metric $D_{\text{ICL}}$, which is simply defined as the average Euclidean distance between each newly generated candidate point $x_i$ and the set of historical points $x_j$ provided as context:

$$D_{\text{ICL}} = \frac{1}{M} \sum_{i=1}^{M} \min_{x_j \in \mathcal{C}} \|x_i - x_j\|_2,$$

where $\mathcal{C}$ is the set of in-context examples shown to the model. A decreasing $D_{\text{ICL}}$ indicates that the model is re-sampling increasingly closer variants of earlier points, i.e., a collapse of the proposal distribution toward previously seen optima. In Figure 15, we illustrate different instances of this metric and how it is computed.

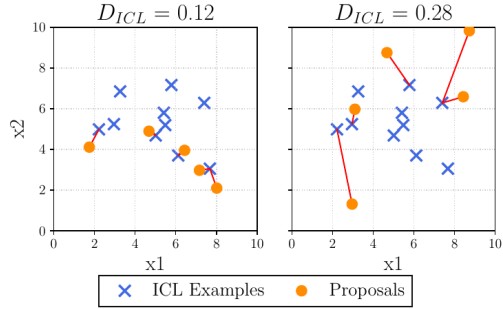

Figure 15: **ICL Divergence Illustration:** An illustration of two distinct search behaviors as measured by the ICL Divergence ($D_{\text{ICL}}$). The left panel shows an exploitative search with a low $D_{\text{ICL}}$ score, while the right panel shows an exploratory search with a high $D_{\text{ICL}}$ score. The score is the average distance (red lines) from each proposal to its nearest ICL example.

**Results.**  Figure 16 shows the evolution of $D_{\text{ICL}}$ on the real-world benchmarks from the main paper. Across all tasks, the global LLM baseline exhibits a rapid decay of $D_{\text{ICL}}$ toward zero. This confirms that the vanilla LLM increasingly produces near-duplicate points around previously observed high-value candidates. In contrast, HOLLM maintains a substantially higher $D_{\text{ICL}}$ throughout the whole optimization. The KD-tree partitioning prevents global collapse by constraining each LLM query to a specific region of the search space. Combined with global context conditioning and bandit-driven region selection, this structure encourages *local refinement* without allowing samples to converge prematurely across the entire domain.

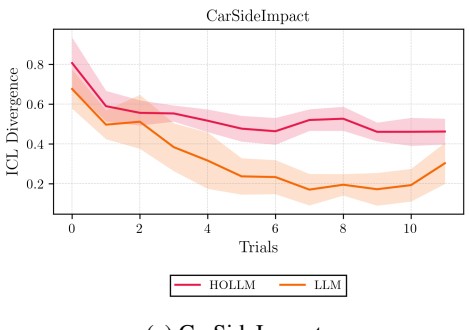

(a) CarSideImpact

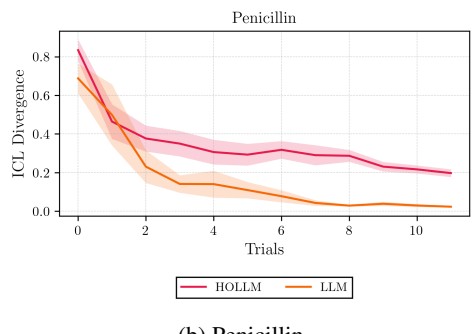

(b) Penicillin

Figure 16: $D_{\text{ICL}}$ over trials on the a) CarSideImpact and b) Penicillin benchmarks. The global LLM baseline rapidly collapses toward zero diversity, while HOLLM maintains broad exploration.

## E   ABLATION STUDY: SCORING, SELECTION, AND SAMPLING

To reinforce the results shown in Figure 8b in the main paper, in Figure 17, we show that consistent trends were observed in preliminary experiments on *NAS-Bench-201* using Gemini-1.5-Flash, confirming that the UCB-V formulation is robust across different problems. In the same figure, we also compare HOLLM against two variants: 1) **KD-Tree + RS**: The LLM sampler is replaced with uniform random samples within the selected subregion. 2) **KD-Tree + GP**: The LLM sampler is replaced by a Gaussian Process (GP) Surrogate which uses the partition history to fit a local model, followed by Expected Improvement (EI) maximization within the selected subregion. We can see that HOLLM significantly outperforms both variants, reconfirms that the LLM's in-context learning ability, when localized, provides a unique and substantial benefit over both a pure exploratory technique (RS) and a standard, highly competitive local BO surrogate (GP-EI).

## F   SURROGATE CALIBRATION ANALYSIS (LLM VS. GP VS. TABPFN)

A key component of HOLLM is using the LLM not just as a candidate sampler, but as a surrogate model too to predict function values and rank candidates within a region. To validate this design choice, we compared the LLM's predictive calibration against Gaussian Processes (GPs) and the recent TabPFN tabular foundation model (Hollmann et al., 2025). In this experiment, we maintained the LLM candidate sampler and only replaced the surrogate model with the aforementioned models:

1. **HOLLM + GP:** Uses the mean function value of the GP as the predicted function value. The training data is the same as the in-context examples given to HOLLM.

2. **HOLLM + TabPFN:** Uses TabPFN (Hollmann et al., 2025), a Transformer-based foundation model pretrained entirely on synthetic tabular data. TabPFN provides quantile predictions and in our case, we use the median as our predicted function value.

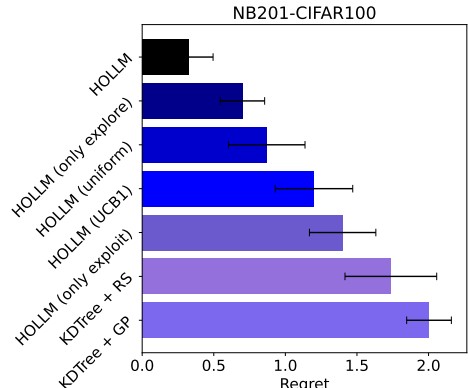

Figure 17: Ablation of the Scoring on the NAS-Bench-201, CIFAR-100 neural architecture search task. The results show the mean and standard error of 6 repetitions. KDTree+RS and KDTree+GP replace the LLM surrogate and candidate sampler with RS and GPs, respectively.

Figure 18 shows the Quantile-Quantile (Q-Q) plots of predicted values vs. ground truth on the real-world benchmarks. These results were directly taken from our follow-up paper on Multi-objective HOLLM (Schwanke et al., 2026). The LLM surrogate (Red) consistently aligns with the diagonal, indicating robust calibration. GPs (Blue) tend to underestimate the spread on Penicillin, while TabPFN (Green) shows significant misalignment, suggesting its prior is less suited for these continuous engineering landscapes. In Table 5 we also report the regression ($R^2$, MSE) and ranking (Kendall $\tau$, Spearman $\rho$) metrics for this experiment, computed using 700 data points in total. The LLM achieves significantly higher ranking correlations (e.g., $\tau \approx 0.76$ on VehicleSafety) compared to GPs ($\tau \approx 0.71$) and TabPFN ($\tau \approx 0.29$). This confirms that the LLM's meta-learned prior provides a superior signal for ranking proposals in sparse, localized regions.

## G   COMPUTATIONAL COMPLEXITY ANALYSIS

Due to the high inference time of LLMs, the primary bottleneck in the pipeline is the $M$ sequential LLM requests per iteration. Since both HOLLM and the global LLM baseline condition on the full history of $t$ points, the context length is effectively identical. Consequently, the attention mechanism cost $\mathcal{O}(t^2)$ (or the associated API latency) is equivalent for both methods. HOLLM only adds a negligible constant number of tokens to the prompt to define the local region boundaries. As shown in Table 6, the partitioning and scoring mechanisms introduce a negligible overhead compared to the LLM inference time. HOLLM requires the same number of inference steps as the global LLM baseline, since they both generate the same number of new points per iteration.

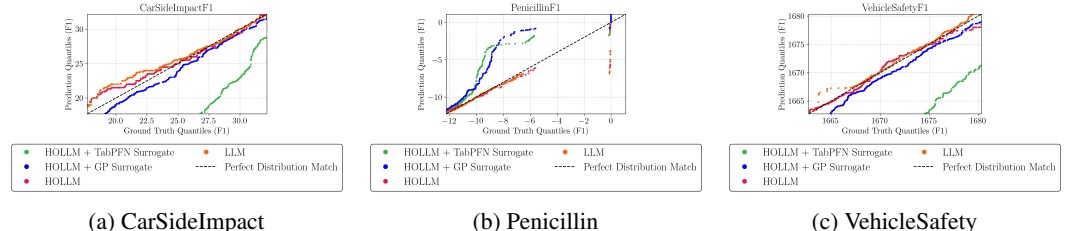

(a) CarSideImpact       (b) Penicillin       (c) VehicleSafety

Figure 18: **Q-Q Plots of Surrogate Calibration.** We compare the predictive distribution of the HOLLM LLM surrogate (Red) against substituting it with a Gaussian Process (Blue) or TabPFN (Green). The LLM consistently aligns closest to the diagonal, indicating superior few-shot calibration.

Table 5: **Quantitative Analysis of Surrogate Performance.** We report regression metrics and ranking correlations for the Penicillin, Vehicle Safety, and Car Side Impact tasks. The LLM consistently outperforms GP and TabPFN baselines in ranking capability (Kendall $\tau$, Spearman $\rho$) and regression accuracy ($R^2$).

| Task | Method | $R^2$ | MSE | Kendall $\tau$ | Pearson $r$ | Spearman $\rho$ | N |
|---|---|---|---|---|---|---|---|
| **Penicillin** | HOLLM (Ours) | 0.742 | 8.06 | 0.684 | 0.868 | 0.851 | 700 |
| | HOLLM + GP | -9.721 | 314.59 | 0.274 | -0.075 | 0.371 | 700 |
| | HOLLM + TabPFN | -173.697 | 5184.01 | 0.375 | -0.138 | 0.487 | 700 |
| | LLM (Global) | 0.756 | 6.20 | 0.730 | 0.875 | 0.854 | 700 |
| **Vehicle Safety** | HOLLM (Ours) | 0.806 | 7.57 | 0.758 | 0.899 | 0.903 | 700 |
| | HOLLM + GP | -9038.058 | 402521.05 | 0.709 | -0.014 | 0.719 | 700 |
| | HOLLM + TabPFN | -16837.928 | 1272547.79 | 0.286 | -0.011 | 0.306 | 700 |
| | LLM (Global) | 0.874 | 7.21 | 0.869 | 0.936 | 0.955 | 700 |
| **Car Side Impact** | HOLLM (Ours) | 0.877 | 5.53 | 0.786 | 0.938 | 0.931 | 700 |
| | HOLLM + GP | 0.010 | 43.35 | 0.781 | 0.679 | 0.802 | 700 |
| | HOLLM + TabPFN | -5.616 | 242.86 | 0.061 | 0.176 | 0.037 | 700 |
| | LLM (Global) | 0.807 | 6.35 | 0.745 | 0.901 | 0.887 | 700 |

Furthermore, we highlight that this approach offers advantageous asymptotic scaling compared to standard Bayesian Optimization. Exact Gaussian Process (GP) inference requires Cholesky decomposition with cubic complexity $\mathcal{O}(t^3)$. In contrast, the Transformer attention mechanism scales quadratically $\mathcal{O}(t^2)$. While LLMs exhibit a higher constant latency factor, this difference implies that HOLLM is asymptotically more computationally efficient than exact GP-based methods as the optimization horizon $t$ grows.

Table 6: **Complexity Analysis of HOLLM, Global LLM, and GP Surrogates**. $t$ denotes the total number of observed points, $L$ the number of leaf regions (with $L \approx t/K$), and $M$ the number of candidate points generated per iteration.

| Component | HOLLM | Global LLM | GP Surrogate |
|---|---|---|---|
| KD-Tree Maintenance | $\mathcal{O}(t \log t)$ | — | — |
| Region Scoring/Selection | $\mathcal{O}(L)$ | — | — |
| Surrogate Update | — | — | $\mathcal{O}(t^3)$ |
| Query / Sampling Cost | $\mathcal{O}(Mt^2)$ | $\mathcal{O}(Mt^2)$ | $\mathcal{O}(t^2)$ |
| Relative Cost | Dominated by LLM inference | Dominated by LLM inference | High for large $t$ |

# H   IMPACT OF LLM CAPABILITY ON DIFFERENT DOMAINS

In the main text, we demonstrated on the *Vehicle Safety* task that HOLLM exhibits high robustness to the underlying LLM's capability, performing well even with smaller models (e.g., Llama-3-8B) where the global LLM baseline fails. In this section, we extended this ablation to two additional benchmarks: *CarSideImpact* (Figure 19a) and *FCNet-Parkinsons* (Figure 19b).

On CarSideImpact, regardless of the LLM model, HOLLM improves on the global LLM baseline. These experiments confirm that HOLLM, via its internal subroutines, reduces the reasoning com-

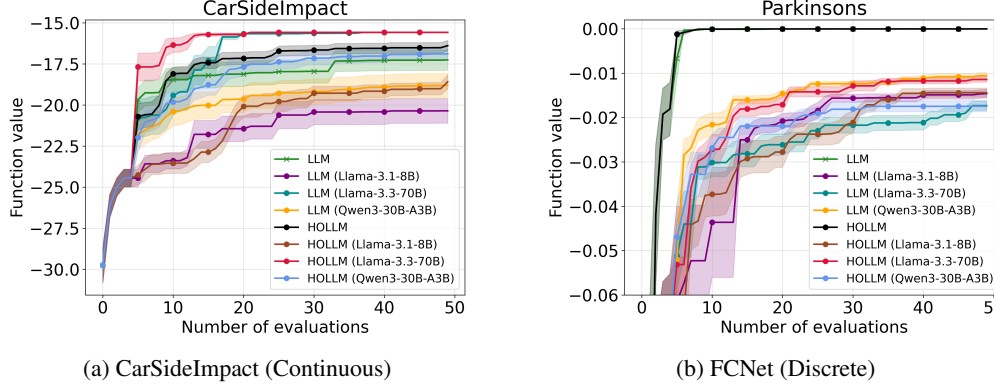

(a) CarSideImpact (Continuous)            (b) FCNet (Discrete)

Figure 19: **Performance across LLM Capabilities.** We compare HOLLM vs. global LLM using various LLM models.

plexity required from the LLM, enabling efficient models (e.g., Llama-3.1-8B) to solve tasks that typically require significantly larger compute budgets.

On the *FCNet-Parkinsons* hyperparameter optimization task, we observe a distinct "ceiling effect" with Gemini-2.0-Flash, which solves the task almost immediately. With the open-weights models we observe distinct behaviors between different LLM models that highlight the complexity of solving discrete HPO using LLM-based optimization. This comparison demonstrates that while HOLLM is universally beneficial in continuous domains (where the KD-Tree partitioning is straightforward), its interaction with discrete landscapes is more dependent on the LLM type rather than the search heuristics.

# I PROMPTS

We design structured prompts to LLMs for both candidate generation and evaluation prediction within our optimization framework. Our prompting strategy consists of two complementary components: a candidate generation prompt (Listing 3) that produces new solutions based on historical observations, and an evaluation prediction prompt (Listing 4) that estimates the quality of proposed candidates, or both in a single prompt. Each prompt follows a systematic structure:

1. **Task specification:** We provide a description of the optimization objective and establish the problem context, including the nature of the search space and optimization goals.

2. **Dynamic constraints:** We define the feasible region constraints (boundaries) derived from the current KD tree partitioning. These constraints are automatically computed based on the selected leaf nodes and translated into natural language descriptions that specify valid input ranges alongside example-evaluation pairs.

3. **In-context examples:** We supply the model with historical observations consisting of previously evaluated points and their corresponding objective function values. These examples serve as demonstrations to guide the model's understanding of the optimization landscape and desired output format. Using global history is necessary for two reasons: 1) *Better prediction in sparse regions*: New or small regions often contain too few points for effective in-context learning. Global history ensures the LLM always has sufficient examples to reason about the function's general behavior; 2) *Robustness*: Global points act as anchors, preventing the LLM from 'overfitting' to local patterns or misinterpreting a local maximum as a global one.

4. **Task-specific instructions:** We provide explicit directives tailored to each prompt's purpose. The candidate generation prompt instructs the model to *propose new configurations*, while the evaluation prediction prompt directs it to *estimate performance* for given candidates. Both prompts enforce structured JSON output formatting for automated parsing of model responses.

Throughout our prompts, we employ placeholder variables denoted with *$* symbols to represent task-specific information that is dynamically populated during execution. Comprehensive examples

and descriptions for each placeholder are provided in Table 7. For the NAS-Bench-201 experiments, we adopt a streamlined approach using a unified prompt structure (Listing 5) that simultaneously elicits both candidate proposals and their predicted evaluations in a single model query, reducing the computational overhead of separate generation and prediction phases.

Table 7: Description of placeholders for candidate proposal and prediction prompts.

| Placeholder | Description | Example of Replaced Text |
|---|---|---|
| `$metrics` | The performance metrics for the specific task. | `F1 (lower is better)` |
| `$region_constraint s` | The allowable ranges or discrete values for the parameters in the configuration search space. | `{`
` lr: range(float([0.0,`
`0.9])),`
` activation:`
`choice(["relu", "tanh"]),`
` num_layer: range(int([1,`
`20]))`
` ...`
`}` |
| `$Region_ICL_exampl es` | Examples of previously evaluated configurations and their performance metrics. These are the in-context learning examples. | `{`
`  {lr: 0,4, activation:`
`"relu", num_layer:`
`8,...}`
`  F1: 5.65`
`  {lr: 0.03, activation:`
`"tanh", num_layer:`
`8,...}`
`  F1: 3.23`
` ...`
`}` |
| `$target_number_of_ candidates` | The number of new configurations that the candidate sampler should generate. | `15` |
| `$candidate_sampler _response_format` | The required JSON structure for each new candidate configuration proposed by the sampler. | `{`
` lr: ?,`
` activation: ?,`
` num_layer: ?`
` ...`
`}` |
| `$target_architectu res` | The set of new configurations for which the surrogate model predicts the performance metrics. | `{`
`  1: {lr: 0,4,`
`activation: "relu",`
`num_layer: 8,...}`
`  2: {lr: 0.03,`
`activation: "tanh",`
`num_layer: 8,...}`
` ...`
`}` |
| `$surrogate_model_r esponse_format` | The required JSON structure for the performance prediction output. | `{F1: ? }` |

```
Suggest 100 random samples for 8 dimensions within the specified bounding
    box with a maximum of 3 decimal places.

 Bounding Box:
   x1_min: 0, x1_max: 1
   x2_min: 0, x2_max: 1
   x3_min: 0, x3_max: 1
   x4_min: 0, x4_max: 1
   x5_min: 0, x5_max: 1
   x6_min: 0, x6_max: 1
   x7_min: 0, x7_max: 1
   x8_min: 0, x8_max: 1

Return the suggestions in the following JSON format exactly, without any
    additional text:
[{"x1": float, "x2": float, "x3": float, "x4": float, "x5": float, "x6":
    float, "x7": float, "x8": float}]
```

Listing 1: Prompt for LLMs simulating 100 8-D uniform random samples.

```
Suggest 80 sample points in 2 dimensions within the specified bounding
    box.

Bounding Box:
  x1_min: 0,  x1_max: 1
  x2_min: 0,  x2_max: 1

such that they are clustered around the points given below.
Points:
  Point 1: x1: 0.25, x2: 0.25
  Point 2: x1: 0.75, x2: 0.75

Return the suggestions in the following JSON format exactly, without any
    additional text:
[{"x1": float, "x2": float}]
```

Listing 2: Prompt for LLMs sampling 80 points around the minima.

```
# Optimization task

## Problem Description
You are tasked with solving a optimization problem that requires finding
    optimal solutions.

- **Evaluation**: Configurations are measured by $metrics

## Constraints
The allowable ranges for the hyperparameters are:
$region_constraints

## Previously Evaluated Architectures
Below are examples of architectures that have been evaluated, showing
    their operations and performance metrics:

$Region_ICL_examples

## Your Task
Generate $target_number_of_candidates new architecture configurations
    that:
1. Are likely to achieve lower $metrics than the examples
2. Are different from all previously evaluated architectures
3. Satisfy all the specified constraints: $region_constraints
```

```
When the region has negative constraints, make sure to take this into
    account when proposing a candidate value.
Before providing the final output, you MUST perform a check to ensure
    every single parameter in your proposed configurations strictly
    adheres to the constraints specified.
Any configuration with a value outside its valid range is incorrect and
    will be rejected.

## Output Format
Each configuration has to follow this format:

$candidate_sampler_response_format

Provide your response in a JSON list containing each proposed
    configuration.
Return only the required JSON list output without additional text.
```

Listing 3: Prompt template used for candidate points generation in the LLM_Generate function.

```
# Configuration Performance Prediction

## Problem Description
You are tasked with predicting the performance of configurations.

- **Evaluation Metric**: $metrics (to be predicted)
- **Constraint**: The allowable ranges for the hyperparameter are:
    $Region_ICL_examples

## Reference configurations with Known Performance
Below are examples of configurations that have been evaluated, showing
    their operations and performance metrics:

## Candidate configurations to Evaluate
You must predict performance for these new configurations:

$target_architectures

## Your Task
1. Predict the $metrics value for each candidate configuration
2. Base your predictions on patterns in the reference examples

## Output Format
Each evaluation has to follow this format:

$surrogate_model_response_format

Provide your response in a JSON list containing each proposed evaluation.
Return only the required JSON list output without additional text.
```

Listing 4: Prompt template used for performance prediction in the LLM_Generate function.

```
Suggest 8 new candidate point(s) for maximizing a blackbox function in a
    6-dimensional search space.

Below are some examples of previously evaluated points with their
    corresponding function values:
[
    {
        "x1": 0.034,
        "x2": 0.287,
        "x3": 0.773,
        "x4": 0.175,
        "x5": 0.755,
        "x6": 0.608,
```

```
            "value": -37.093
    },
    {
        "x1": 0.199,
        "x2": 0.433,
        "x3": 0.405,
        "x4": 0.779,
        "x5": 0.186,
        "x6": 0.594,
        "value": -37.84
    },
    {
        "x1": 0.447,
        "x2": 0.342,
        "x3": 0.97,
        "x4": 0.087,
        "x5": 0.115,
        "x6": 0.533,
        "value": -44.52
    },
    {
        "x1": 0.949,
        "x2": 0.127,
        "x3": 0.659,
        "x4": 0.546,
        "x5": 0.049,
        "x6": 0.265,
        "value": -33.067
    }
]

The search space is defined by the following bounding boxes:
    x1_min: 0.492, x1_max: 1.000
    x2_min: 0.000, x2_max: 1.000
    x3_min: 0.000, x3_max: 1.000
    x4_min: 0.000, x4_max: 1.000
    x5_min: 0.000, x5_max: 1.000
    x6_min: 0.000, x6_max: 1.000

Based on the examples above, suggest candidate points that balance
    exploration (sampling new regions) with exploitation (focusing on
    promising areas where function values are good). Each candidate point
     must lie within the specified bounding boxes. In addition, predict
    an estimated function value for each candidate.

Return the suggestions in the following JSON format exactly, without any
    additional text:
[{"x1": float, "x2": float, "x3": float, "x4": float, "x5": float, "x6":
    float, "value": float}]
```

Listing 5: Prompt example used for simultaneous candidate generation and performance prediction in the LLM_Generate function for NAS-Bench-201.

