# OpenReview forum: "Improving LLM-based Global Optimization with Search Space Partitioning"
_ICLR.cc/2026/Conference — ICLR 2026 Poster_

### Official Review · Reviewer_LeUX · 2025-10-22

**Soundness:** 2
**Presentation:** 3
**Contribution:** 2
**Rating:** 4
**Confidence:** 3

**Summary:**

The paper addresses black-box global optimization with LLMs and argues that asking an LLM to propose points globally often yields sparse, biased coverage—especially in higher dimensions or without domain priors. It introduces HOLLM, a loop that (i) adaptively partitions the search space with a KD-tree, (ii) treats each leaf as a meta-arm, (iii) scores leaves by combining best observed improvement, geometric size, and a variance-aware exploration bonus, and (iv) stochastically selects regions before prompting the LLM to generate local candidates, which are then evaluated and fed back. The method re-fits the KD-tree each round to avoid premature localization and uses a cosine-style schedule to front-load exploration. Experiments span continuous synthetic benchmarks (Hartmann, Rosenbrock, Rastrigin, Lévy), FCNet-9D hyperparameter optimization on four datasets, and three real-world continuous tasks (Penicillin, Vehicle Safety, Car Side Impact). HOLLM typically matches or shows modest gains over BO/ES and global-LLM baselines, but it is not consistently dominant.

**Strengths:**

1. Methodological novelty with a clean, coherent design. The paper marries KD-tree–based adaptive space partitioning with a bandit-style composite scoring rule—combining best observed improvement, geometric volume, and a variance-aware (UCB-V–like) exploration term—and then prompts the LLM to generate candidates locally within selected regions. Re-fitting the partition each round helps prevent premature commitment and keeps the search adaptive. Overall, the loop (partition, score, select, local LLM proposals, evaluate) is conceptually simple yet technically sound.
2. Across diverse tasks, HOLLM’s trajectories are comparatively stable against strong baselines. Notably, on many settings the global-LLM variant tends to stall, whereas HOLLM continues to improve. This is likely because “partition + scoring + local sampling” mitigates sampling bias in LLM-guided optimization and reduces brittleness to model idiosyncrasies.
3. Wide experimental coverage with appropriate baselines. The evaluation spans continuous synthetic functions, FCNet-9D hyperparameter optimization, and three real-world continuous tasks, and it compares against a solid suite of methods: BO (e.g., CQR, GP-EI), evolutionary strategies, random search, a carefully controlled global-LLM baseline, and RS+KD-Tree.
4. Component and hyperparameter ablations that aid interpretation. Both the main text and appendix report ablations/visualizations for key knobs, such as the exploration annealing schedule (α), batch/selection sizes (b/M/k), and leaf capacity. This helps clarifying how each component contributes to performance and stability.

**Weaknesses:**

1. Average gains are modest. On continuous tasks, HOLLM is clearly best only on a subset (Lévy, Rosenbrock, Car Side Impact, Vehicle Safety). On the other continuous benchmarks, there exist methods that outperform it by a visible margin; several curves overlap within error bars, so the advantage is not consistent. On FCNet-9D (discrete HPO), HOLLM and the global-LLM baseline are essentially overlapping—partitioning brings little benefit in this regime. Because the paper does not provide theory, the empirical results carry the core argumentative burden. Since improvements on continuous tasks are modest, and those on FCNet are near zero, it’s hard to argue broad superiority; the main benefit appears to be robustness rather than overall performance.
2. LLM ablation is under-scoped. The multi-LLM comparison appears only on a single task (Vehicle Safety). To demonstrate the practicality of the algorithm, the paper should repeat the ablation across multiple continuous benchmarks.

**Questions:**

1. Experimental results are not particularly strong in my view. Based on the experimental evidence, can you provide a detailed, results-driven explanation of what the algorithm actually contributes, with pointers to the specific tasks and figures?
2. The multi-LLM ablation is only on Vehicle Safety. Can you repeat it on more tasks (1 additional continuous benchmarks and 1 discrete task)?
I would consider increasing my score if you both (i) provide a detailed, results-grounded explanation of the algorithm’s contributions (answering my first question), and (ii) expand the LLM ablation to more tasks.

---

> ### Author Response · Authors · 2025-11-20
>
> Thank you for the detailed review and valuable feedback. We appreciate your critical assessment of the performance margins. Below, we address your main concern.
>
> ---
>
> >***Experimental results are not particularly strong in my view. Based on the experimental evidence, can you provide a detailed, results-driven explanation of what the algorithm actually contributes, with pointers to the specific tasks and figures?***
>
> We show that HOLLM provides the most value in:
> 1. **Optimizing complex real-world continuous functions**. The benchmarks that you mentioned (Lévy, Rosenbrock, Car Side Impact, and Vehicle Safety) are some of the most complex (with high multimodality, ill-conditioned, etc.) and difficult to optimize in our benchmark suite. For instance, *Car Side Impact* and *Vehicle Safety* represent non-smooth, high-stakes engineering problems. HOLLM’s ability to exploit local structure yields significant gains here, which matters a lot in practice, especially when single evaluations take several hours/days. On the other hand, synthetic functions such as *Rosenbrock* punish myopic greedy search. HOLLM’s partitioning provides a mitigation strategy for the potential early convergence seen in the global LLM optimizer.
> 2. **Enabling efficient optimization with smaller LLMs.** While the gap is moderate when using larger models (like Llama-3.3-70B), we have added new empirical evidence (Figure 20 in Appendix I of the revision) that shows that the performance gains on the real world tasks are relatively higher when using smaller, efficient models (e.g., Llama-3.1-8B). The global LLM baseline performance degrades severely because the model struggles to reason over the complex, global history when using these smaller LLMs. HOLLM enables these smaller models to perform on par with HOLLM and global LLM baselines using significantly larger models. By decomposing the global problem into simpler local sub-problems, HOLLM reduces the reasoning burden on the LLM. This makes HOLLM a vital contribution for *efficient BBO*, allowing users to deploy cheaper, faster, local LLMs while achieving results that previously required expensive proprietary APIs.
>
> ---
>
> >***Performance on Discrete Tasks (FCNet)***
>
> We agree that gains on discrete HPO tasks (FCNet) appear marginal or overlapping. Two factors contribute to this:
> 1. **Scale of Objectives:** The absolute error range for these benchmarks is extremely small (order of $10^{-4}$ to $10^{-5}$) for the function being optimized (MSE). Overlap in the plots often represents differences that are negligible in practice for HPO, as both methods are close to the global optimum.
> 2. **Discrete Space:** As you also mentioned, this experiment on FCNet was mainly to demonstrate the robustness of HOLLM by applying it on a discrete search space (we chose FCNet since it was already inside SyneTune). Discrete hyperparameter spaces often lack the continuous local geometry that HOLLM’s partitioning strategy exploits. Nevertheless, the fact that HOLLM is better or on par compared to the baseline confirms its robustness even when partitioning does not offer a direct geometric advantage.
>
> ---
>
> >***The multi-LLM ablation is only on Vehicle Safety. Can you repeat it on more tasks (1 additional continuous benchmarks and 1 discrete task)?***
>
> We have performed the requested multi-LLM ablation on **CarSideImpact** (additional continuous task) and **FCNet** (discrete task). The results (added to Appendix I, Figure 20/a, of the revision) are consistent with the findings on *Vehicle Safety*: Regardless of the LLM model, HOLLM improves on the global LLM baseline. On CarSideImpact, while both methods perform well with the 70B Llama model, the global LLM baseline performance drops significantly when switching to the 8B Llama model. In contrast, HOLLM with Llama-3-8B shows improved performance. These results on real-world tasks demonstrate that HOLLM, through its subroutines, allows smaller, more efficient LLMs to achieve comparable results to much larger LLMs in global optimization, a crucial advantage for deployment in resource-constrained environments. On FCNet-Parkinsons (Figure 20/b), which is a discrete hyperparameter space, Gemini-2.0-Flash model solves the task almost instantly (reaching 0.0 MSE), rendering the search method less relevant for this specific LLM. However, open-weights models seem to struggle with this task, indicating that the meta-learned prior during LLM pretraining plays a crucial role for these tasks.
>
> ---
>
> Thank you again for your valuable feedback and the additional experiments that helped make our paper stronger. We are happy to engage in further discussion regarding any remaining concerns. Otherwise, we hope that you will consider raising your score.

---

> > ### Comment · Reviewer_LeUX · 2025-11-21
> >
> > Your paper does have some practical value, and the algorithm itself has a certain degree of innovation. I have already raised my score.

---

> > > ### Author Response · Authors · 2025-11-22
> > >
> > > Thank you very much for taking the time to reevaluate our work and for raising your score. We appreciate your recognition of both the practical value and the methodological contributions of HOLLM. If there is anything further that you feel would make the paper even stronger or more convincing, we would be grateful for your guidance and are happy to incorporate additional clarifications or analyses.

---

### Official Review · Reviewer_Y2NU · 2025-10-27

**Soundness:** 3
**Presentation:** 3
**Contribution:** 2
**Rating:** 6
**Confidence:** 4

**Summary:**

- This paper introduces HOLLM, a global BBO algorithm that uses adaptive search space partitioning with LLM-based hypothesis proposal (within selected subregions) and evaluation (global)
- It uses a bandit-style metric at the partition-level to trade off exploitation, geometric coverage, and statistical uncertainty
- The algorithm follows the process of (1) partitioning the space, (2) ranking leaf regions, (3) sampling from within the regions of selected partitions, (4) using an LLM to propose local points, (5) evaluating and selecting the best proposals

**Strengths:**

- The paper gives an intuitive and empirical evidence of LLM failure modes in high-dimensional spaces (biased coverage, mode-seeking), which it addresses through searching/sampling within bounded subregions
- The algorithm makes sense, combining KD-tree partitioning, UCB-style scoring/selection over the partitions (as arms), and LLM-buided local BO. While each of those components exist in some form in BO/MAB/hierarchical bandit literature, the particular design feels suitable and well-motivated
- The arm-level scoring function and stochastic selection policy is novel (AFAIK), based on best-observed improvement (exploitation), normalized HV (geometric exploration), and UCB-V (uncertainty)

**Weaknesses:**

- The arm-level scoring functions feels slightly ad-hoc and slightly complicated, it is not clear why the HV-based term and UCB-V (as opposed to other UCB variants) is necessary and used. The HOLLM vs global LLM-based BO is a good ablation, but it would also be interesting to isolate the different terms (e.g., UCB-variance) to see if they matter in practice.
- It would be interesting to understand the LLM performance specifically, e.g., how well the surrogate performance is vs a GP
- One possible limitation of this work is that even with KD-tree partitioning, this might degrade in higher-dimensional spaces. Based on my understanding, the paper's benchmarks are mainly < 10D, it is not clear whether this partitioning will give proportional empirical improvements (say in 50D) where TURBO-style trust regions or CMA_ES covariance structures might be more robust
- It is not clear to me why the LLM sampling/surrogate sees the global results, and not just region-specific results
- Minor: the cosine annealing for exploration is sensible, but we don't see quantitative ablations around sensitivity.
- Minor: how are the number of partition regions selected (I might have missed this), it feels like an important hyperparameter.
- Minor: in the results section, it is a bit confusing to interpret (with maximize vs minimize), it would be hlepful to standardize sign convention and introduce visual hints
- Minor: this is not a major concern in my mind, but as the work draws parallels to HOO/bandit-style UCB-V, it is worth pointing out that this field of literature cares particularly about regret/concentration analysis.

**Questions:**

- The LLM surrogate does not include any uncertainty. How good is the mean prediction, especially with LLMs of different capabilities
- Can the authors clarify the computational overhead (e.g., wall-clock time) for each aspect of their pipeline (e.g., dynamic re-partitioning, local LLM-based BO)
- It would be interesting, but not a priority, to see results on benchmarks with more noise and understanding how the partitioning + variance-sensitive scoring function performs
- How limiting are the axis-aligned partitions? Would this be applicable to different problems and where might it be a bottleneck?

---

> ### Author Response · Authors · 2025-11-20
> **Official Comment by Authors (Part 1/3)**
>
> Thank you for your thorough review, your constructive critique and the encouraging score. We address your primary concerns below.
>
> ---
>
>
> >***It would be interesting to understand the LLM performance specifically, e.g., how well the surrogate performance is vs a GP.*** and ***How good is the mean prediction, especially with LLMs of different capabilities***
>
> We agree that a direct comparison against standard, simple samplers within the HOLLM framework is crucial to justify the LLM choice. We had already conducted an ablation study on the NB201-C100 benchmark comparing HOLLM against two variants:
> - **KD-Tree + RS**: The LLM sampler is replaced with uniform random samples within the selected subregion.
> - **KD-Tree + GP**: The LLM sampler is replaced by a Gaussian Process (GP) Surrogate which uses the partition history to fit a local model, followed by Expected Improvement (EI) maximization within the selected subregion.
>
> Referring again to Figure 18 in Appendix F, we can see that HOLLM significantly outperforms both variants, reconfirms that the LLM's in-context learning ability, when localized, provides a unique and substantial benefit over both a pure exploratory technique (RS) and a standard, highly competitive local BO surrogate (GP-EI).
>
> Regarding the surrogate prediction quality, our new quantitative analysis (included in Appendix G of the revision) demonstrates that the LLM surrogate in HOLLM significantly outperforms baselines like Gaussian Processes (GPs) and the recent state-of-the-art tabular foundation model, TabPFN (Hollman et al. 2025), in both calibration (Figure 19) and ranking (Table 6). On real-world tasks such as VehicleSafety and CarSideImpact, the LLM achieves high predictive fidelity ($R^2$≈0.80-0.87) and strong rank correlations (Kendall $\tau$≈0.76-0.78). Additionally, our Q-Q plot analysis reveals that the LLM’s predictions align closely with the ground truth diagonal, whereas GPs tend to underestimate variance sometimes and TabPFN shows significant misalignment.
>
> | Task | Method | R² | MSE | Kendall $\tau$ | Pearson R | Spearman $\rho$ | N |
> | :--- | :--- | :--- | :--- | :--- | :--- | :--- | :--- |
> | **Penicillin** | HOLLM (Ours) | 0.742 | 8.06 | 0.684 | 0.868 | 0.851 | 700 |
> |  | HOLLM + GP Surrogate | -9.721 | 314.59 | 0.274 | -0.075 | 0.371 | 700 |
> | | HOLLM + TabPFN Surrogate | -173.697 | 5184.01 | 0.375 | -0.138 | 0.487 | 700 |
> | | LLM (Global) | 0.756 | 6.20 | 0.730 | 0.875 | 0.854 | 700 |
> | **Vehicle Safety** | HOLLM (Ours) | 0.806 | 7.57 | 0.758 | 0.899 | 0.903 | 700 |
> |  | HOLLM + GP Surrogate | -9038.058 | 402521.05 | 0.709 | -0.014 | 0.719 | 700 |
> |  | HOLLM + TabPFN Surrogate | -16837.928 | 1272547.79 | 0.286 | -0.011 | 0.306 | 700 |
> |  | LLM (Global) | 0.874 | 7.21 | 0.869 | 0.936 | 0.955 | 700 |
> | **Car Side Impact** | HOLLM (Ours) | 0.877 | 5.53 | 0.786 | 0.938 | 0.931 | 700 |
> |  | HOLLM + GP Surrogate | 0.010 | 43.35 | 0.781 | 0.679 | 0.802 | 700 |
> |  | HOLLM + TabPFN Surrogate | -5.616 | 242.86 | 0.061 | 0.176 | 0.037 | 700 |
> |  | LLM (Global) | 0.807 | 6.35 | 0.745 | 0.901 | 0.887 | 700 |
>
> ---
>
> >***One possible limitation of this work is that even with KD-tree partitioning, this might degrade in higher-dimensional spaces. Based on my understanding, the paper's benchmarks are mainly < 10D, it is not clear whether this partitioning will give proportional empirical improvements (say in 50D) where TURBO-style trust regions or CMA_ES covariance structures might be more robust***
>
> Thank you for raising this point. We acknowledge that HOLLM’s main bottleneck is the LLM inference time, which can become high in high-dimensional spaces. In Table 4 (Appendix C.1.1) we show comparisons to LA-MCTS (with different local models), TuRBO, CMA-ES, SMAC and HEBO on 3 of the Ackley 20D benchmark (last column). We ran each method 5 times using their official implementations and reported mean +/- std. As demonstrated for an evaluation budget of 100 function evaluations, HOLLM is still superior, but even when running LA-MCTS for 1000 function evaluations, it still is not able to find the global minimum in Ackley 20D, while HOLLM does that in less than 50 evaluations. Note that we did not include these baselines in the main paper, since they are not implemented in SyneTune, the framework we used to run all the experiments/methods in the main paper using the same API.

---

> > ### Author Response · Authors · 2025-11-20
> > **Official Comment by Authors (Part 2/3)**
> >
> > >***The arm-level scoring functions feels slightly ad-hoc and slightly complicated, it is not clear why the HV-based term and UCB-V (as opposed to other UCB variants) is necessary and used. The HOLLM vs global LLM-based BO is a good ablation, but it would also be interesting to isolate the different terms (e.g., UCB-variance) to see if they matter in practice.***
> >
> > Thank you for pointing this out. We will clarify it more in detail in the main paper. We acknowledge that the scoring function seems complex, but the terms play necessary, complementary roles:
> > - **HV-based term**: UCB-based terms assign the same score to a tiny sub-region and a massive partition if their sample statistics (mean/variance) and number of samples per region are identical (which is often the case for a small leaf size as in our experiments). The HV term corrects this by biasing selection toward larger, less-dense regions, ensuring that broad exploration (especially during the first iterations) is not penalized.
> > - **UCB-V**: This term is especially critical in later iterations. Tiny regions often exhibit high variance and deserve further sampling even if their mean is currently lower; UCB-V captures this, whereas standard UCB1 tends to under-explore these volatile but promising areas.
> >
> > In preliminary experiments, we already evaluated HOLLM with other UCB terms, more specifically UCB1. In Appendix F, we have added the results of the preliminary experiments on NAS-Bench-201 (Fig. 18) and the new experimental results we obtained on the Vehicle Safety real-world task (Fig. 17). In both cases, we can notice that the replacing UCB-V with UCB1 resulted in worse performance. Furthermore, we also ablated other relevant terms from the scoring function, such as removing the  individual exploration or exploitation terms completely. As shown in the results, removing either term degrades convergence speed and final quality compared to the full HOLLM scoring function (black line). We also tested sampling regions uniformly at random rather than from the distribution parameterized with the region scores. This performed significantly worse in the Vehicle Safety benchmark, *confirming that our selection mechanism is beneficial for guiding the LLM to promising subspaces*.
> >
> > ---
> >
> > >***It is not clear to me why the LLM sampling/surrogate sees the global results, and not just region-specific results.***
> >
> > This is a careful observation from your side. Using global history in HOLLM is necessary for two reasons:
> > 1. *Better prediction in sparse regions*: New or small regions often contain too few points for effective in-context learning. Global history ensures the LLM always has sufficient examples to reason about the function's general behavior;
> > 2. *Robustness*: Global points act as anchors, preventing the LLM from 'overfitting' to local patterns or misinterpreting a local maximum as a global one. In preliminary experiments, we have confirmed the superior performance of HOLLM using the global context. We have added this justification in Appendix D.
> >
> > ---
> >
> > >***Minor: cosine annealing sensitivity and on the number of partition regions selected***
> >
> > We provide sensitivity analysis on the default values we chose for these hyperparameters in Appendix C.2. More specifically, Figure 14 shows the results of varying $\alpha_{max}$ in the cosine annealing schedule and Figure 13 (rightmost plot) on the number of partitions selected.
> >
> > ---
> >
> > >***Minor: standardize sign convention and introduce visual hints***
> >
> > Thank you for this suggestion. We frame the problem as a maximization one. We will standardize the sign in the final revision of our paper to improve readability.
> >
> > ---
> >
> > >***Minor: as the work draws parallels to HOO/bandit-style UCB-V, it is worth pointing out that this field of literature cares particularly about regret/concentration analysis***
> >
> > We agree that theoretical regret analysis is central to the bandit literature and we have already added this point in the “Limitations” paragraph. However, applying such analysis here is quite challenging because LLMs, acting as black-box priors, do not adhere to the strict continuity or smoothness assumptions (e.g., weak Lipschitz conditions) required for standard HOO proofs, unless strongly assuming so. We view HOLLM as an empirical adaptation of the  "Optimism in the Face of Uncertainty" principle as a design heuristic to guide the LLM. While we lack a theoretical regret bound, our experiments provide empirical regret analysis (showing rapid convergence to optima), demonstrating that the mechanism of optimistic exploration works effectively in practice in our case.

---

> > > ### Author Response · Authors · 2025-11-20
> > > **Official Comment by Authors (Part 3/3)**
> > >
> > > >***Can the authors clarify the computational overhead (e.g., wall-clock time) for each aspect of their pipeline (e.g., dynamic re-partitioning, local LLM-based BO)***
> > >
> > > We have analyzed the computational complexity of HOLLM in the newly added Appendix H and Table 7 of the revised paper PDF. Most importantly, due to the high inference time of LLMs, the primary bottleneck in the pipeline is the $M$ sequential LLM requests per iteration. Since HOLLM conditions on the full history of $t$ points (the context length), the attention mechanism computation cost $\mathcal{O}(t^2)$ (or the associated API latency) dominates the partitioning and scoring ones as shown in the table below. For a concrete example, optimizing a 6D continuous problem took approximately 3.5 seconds per iteration.
> > >
> > > | Component | Complexity (per iter.) | Relative Cost |
> > > | :--- | :--- | :--- |
> > > | **KD-Tree Construction** | $\mathcal{O}(t \log t)$ | Negligible |
> > > | **Region Scoring** | $\mathcal{O}(L)$ | Negligible |
> > > | **LLM Inference** | $\mathcal{O}(M t^2)$ | **Dominant** |
> > >
> > > *$t$: total points, $L$: number of leaves, $M$: candidates per iter.*
> > >
> > > ---
> > >
> > > >***It would be interesting, but not a priority, to see results on benchmarks with more noise and understanding how the partitioning + variance-sensitive scoring function performs.***
> > >
> > > We appreciate this point, as the design of the variance-sensitive UCB-V term is intended precisely for noisy environments. We have conducted the requested experiment on the *Vehicle Safety* task by adding **zero-mean Gaussian noise ($\mathcal{N}(0, \sigma^2)$) with a standard deviation ($\sigma$) fixed at $1\%$ of the global objective function range** (following the setup in https://proceedings.neurips.cc/paper/2021/file/11704817e347269b7254e744b5e22dac-Paper.pdf). Results (Figure 21 in Appendix J of the revision) show a larger gap between the performance of HOLLM and the global LLM baseline, as contrasted to the original results on Vehicle Safety without noise. This confirms that the combination of partitioning and variance-aware scoring improves HOLLM’s robustness to noise.
> > >
> > > ---
> > >
> > > >***How limiting are the axis-aligned partitions? Would this be applicable to different problems and where might it be a bottleneck?***
> > >
> > > The use of axis-aligned partitioning limits geometric flexibility, especially for objective functions with strong variable correlations, where diagonal structures may be approximated only through a sequence of increasingly fine rectangular splits. However, in contrast to uniform sampling, which is strictly constrained by box geometry, LLMs can infer such correlations directly from in-context examples and often propose candidates along diagonally structured manifolds even when the bounding region is axis-aligned. Axis-aligned constraints (e.g., $x_1 < 0.4$) are also simpler to compute, more token-efficient, and easier for LLMs to interpret reliably. In contrast, oblique constraints (e.g., $3x_1 + 2x_2 < 0.4$) substantially increase prompt complexity and may introduce additional reasoning challenges for the model.
> > >
> > > In an early version of our method, we also evaluated more flexible partitioning strategies, such as Voronoi diagrams. However, computing geometric volumes for arbitrary non-rectangular cells in high dimensions can get difficult, as such approaches would require approximate volume estimation (e.g., via Monte Carlo sampling), and even determining cell boundaries becomes non-trivial. For these reasons, we decided to use the axis-aligned KD-tree partitions, as they offer the best trade-off between geometric expressiveness, computational tractability, and LLM compatibility.
> > >
> > > Regarding broader applicability, we note two other potential bottlenecks: spaces with many uninformative dimensions can degrade tree efficiency (addressable via feature selection), and non-Euclidean inputs like graphs require mapping to Euclidean embeddings via encoders to be compatible with our partitioning scheme.
> > >
> > > ---
> > >
> > > Thank you again for your valuable feedback and insights, which have ultimately improved our paper with the additional experiments. We are happy to discuss any further concerns you may have during this discussion phase. Otherwise, we would really appreciate it if you would consider raising your score.
> > >
> > > ---
> > >
> > > *-- References --*
> > >
> > > Hollman et al. "Accurate predictions on small data with a tabular foundation model". In Nature 637, 2025

---

### Official Review · Reviewer_dF12 · 2025-10-31

**Soundness:** 2
**Presentation:** 3
**Contribution:** 1
**Rating:** 4
**Confidence:** 5

**Summary:**

This paper addresses a key limitation of using Large Language Models (LLMs) for global optimization: their tendency to sample sparsely and inefficiently in high-dimensional search spaces. The authors propose HOLLM (Hierarchical Optimization with Large Language Models), a novel algorithm that integrates LLM-based candidate sampling with an adaptive search space partitioning strategy. HOLLM iteratively builds a KD-tree to divide the search space into smaller subregions. It then uses a bandit-inspired utility score to select the most promising subregions, effectively balancing exploration (large or uncertain regions) and exploitation (regions with good observed values). An LLM is then prompted to generate new candidate points specifically within these selected, smaller regions. The authors' empirical results on benchmark functions demonstrate that HOLLM matches or outperforms other global optimization methods and significantly improves upon a baseline "global LLM" sampler.

**Strengths:**

- The paper clearly identifies and demonstrates a practical weakness of LLM-based samplers—their high bias and inability to cover a space effectively (as shown in Figure 1)—and proposes an intuitive solution.
- The inclusion of a "global LLM" baseline is crucial, as it provides strong evidence that the partitioning framework itself, not just the use of an LLM, is responsible for the performance gains

**Weaknesses:**

- The paper's primary contribution appears incremental. The core components—hierarchical space partitioning (e.g., KD-trees) and bandit-based region selection—are well-established techniques in the black-box and hierarchical optimization literature (e.g., HOO, MABs). The method seems to primarily substitute a traditional sampler within this framework with an LLM, which may limit the work's fundamental novelty.
- There is a notable disconnect between the paper's motivation and its empirical evaluation. The introduction suggests LLMs could unlock optimization for novel domains (e.g., language-based tasks) where traditional numeric methods are unsuitable. However, the experiments are confined to standard numerical benchmarks (synthetic functions, hyperparameter tuning) where classic methods are already effective. This evaluation fails to substantiate the primary motivating claim for using LLMs.
- The framework's components, apart from the LLM sampler, are standard building blocks in black-box optimization. This reinforces the concern about incremental contribution, as it is unclear why an LLM is a necessary choice. The partitioning and selection mechanism  could seemingly be combined with other advanced samplers (e.g., from PSO, DE, or even a simple Gaussian sampler). The paper lacks a crucial ablation study to isolate and justify the unique benefit of using an LLM over these simpler, well-understood alternatives.
- The results on synthetic benchmarks are difficult to interpret and potentially unconvincing. The authors should explicitly state the known global optima for these functions. There is a concern that some functions may have trivial solutions (e.g., $x^*=0$) that an LLM, due to its inherent biases, might guess easily. This suspicion is heightened by the surprisingly strong performance of the "global LLM" baseline, which reportedly outperforms most numeric methods (Table 4). This counter-intuitive baseline result calls the benchmark's difficulty and validity into question.
- On the hyperparameter optimization tasks, the proposed HOLLM does not show a significant advantage over the simpler "global LLM" baseline. Given that the global method is far less complex, this result questions the practical utility of the added HOLLM framework. The paper needs to provide a cost-benefit analysis, including the computational overhead of the partitioning and scoring components. Furthermore, the work fails to disentangle whether the observed performance (of both HOLLM and the global baseline) is due to the algorithmic structure or simply the LLM's powerful, pre-trained "intrinsic knowledge".

**Questions:**

- What is the computational complexity (e.g., wall-clock time) of the HOLLM framework, specifically the KD-tree construction and region scoring, relative to the (presumably expensive) LLM sampling calls? How does this overhead scale as the number of evaluated points t increases?
- The paper's motivation (Fig. 1) suggests LLMs are bad at sampling (i.e., biased). The method's success implies this bias is now a useful "prior". Could you elaborate on what specific, useful properties this "meta-prior" is assumed to have, especially when constrained to small subregions?

---

> ### Author Response · Authors · 2025-11-20
> **Official Comment by Authors (Part 1/2)**
>
> Thank you for taking the time to review our paper and your constructive critique. We address your primary concerns below.
>
> ---
>
> >***The paper's primary contribution appears incremental***
>
> While we agree that the core components (KD-trees, bandit selection) are already established concepts, our primary contribution lies their novel and effective integration with LLMs for global optimization, specifically to overcome the LLM's inherent sampling biases demonstrated in Figure 1 and now in Appendix E of the revised paper PDF. To support our claim, the other reviewers also acknowledge the novelty aspects of our method. To quote Reviewer Y2NU: “*The algorithm makes sense, combining KD-tree partitioning, UCB-style scoring/selection over the partitions (as arms), and LLM-buided local BO. While each of those components exist in some form in BO/MAB/hierarchical bandit literature, the particular design feels suitable and well-motivated. The arm-level scoring function and stochastic selection policy is novel (AFAIK), based on best-observed improvement (exploitation), normalized HV (geometric exploration), and UCB-V (uncertainty)*” and Reviewer LeUX: “*Methodological novelty with a clean, coherent design. The paper marries KD-tree–based adaptive space partitioning with a bandit-style composite scoring rule—combining best observed improvement, geometric volume, and a variance-aware (UCB-V–like) exploration term—and then prompts the LLM to generate candidates locally within selected regions. Re-fitting the partition each round helps prevent premature commitment and keeps the search adaptive. Overall, the loop (partition, score, select, local LLM proposals, evaluate) is conceptually simple yet technically sound.*”
>
> ---
>
> >***There is a notable disconnect between the paper's motivation and its empirical evaluation. The introduction suggests LLMs could unlock optimization for novel domains (e.g., language-based tasks) where traditional numeric methods are unsuitable. However, the experiments are confined to standard numerical benchmarks (synthetic functions, hyperparameter tuning) where classic methods are already effective. This evaluation fails to substantiate the primary motivating claim for using LLMs.***
>
> We have not made such a claim in the introduction. Can you please point exactly where you found the part where we “suggest LLMs could unlock optimization for novel domains (e.g., language-based tasks) where traditional numeric methods are unsuitable.”?
>
> ---
>
> >***The partitioning and selection mechanism could seemingly be combined with other advanced samplers (e.g., from PSO, DE, or even a simple Gaussian sampler). The paper lacks a crucial ablation study to isolate and justify the unique benefit of using an LLM over these simpler, well-understood alternatives.***
>
> This is a very valid concern. As the reviewer suggested, our algorithm is highly modular and the LLM sampler can be replaced with other non-LLM ones. In preliminary experiments, we had already investigated this using local GP-EI samplers and the latter underperformed significantly. We added these results in our new ablation study (included in Appendix F, Figure 18, and discussed in the response to Reviewer Y2NU), where we compare HOLLM against KD-Tree + RS (random sampler) and KD-Tree + GP-EI (a local Gaussian Process Expected Improvement sampler in each region). As we can see in Figure 18, HOLLM significantly outperforms both, confirming that the performance gain is due to the synergy between the partitioning framework and the localized LLM's sampling capabilities, not just the framework itself. Moreover, in Appendix G, we also assess the predictive performance of the LLM as a surrogate model by comparing it with GPs and the recent TabPFN tabular foundation model.
>
> ---
>
> >***The results on synthetic benchmarks are difficult to interpret and potentially unconvincing. The authors should explicitly state the known global optima for these functions.***
>
> The global minimizers for each of the synthetic functions are in Table 1 in the Appendix. As you suspect, some of them, but not all, have global minimizers at 0, which might make the task easier for the LLM. This seems to be the case for Rastrigin, but not for the other benchmarks, where the global LLM baseline struggles even though the global optimum is at 0. The main problem is that the global LLM might still collapse its proposed solutions to a found local optimum, as demonstrated in the new diversity analysis experiments in Appendix E.

---

> > ### Author Response · Authors · 2025-11-20
> > **Official Comment by Authors (Part 2/2)**
> >
> > >***On the hyperparameter optimization tasks, the proposed HOLLM does not show a significant advantage over the simpler "global LLM" baseline. Given that the global method is far less complex, this result questions the practical utility of the added HOLLM framework.***
> >
> > We agree that gains on the HPO tasks (FCNet) appear not significant. Two factors contribute to this: (1) The absolute error range for some of these benchmarks is extremely small (order of $10^{-4}$ to $10^{-5}$) for the function being optimized (MSE). Overlap in the plots often represents differences that are negligible in practice for HPO, as both methods are close to the global optimum. (2) This experiment on FCNet was mainly to demonstrate the robustness of HOLLM by applying it on a discrete search space (we chose FCNet since it was already inside SyneTune). Discrete hyperparameter spaces often lack the continuous local geometry that HOLLM’s partitioning strategy exploits. Nevertheless, the fact that HOLLM is better or on par compared to the baseline confirms its robustness even when partitioning does not offer a direct geometric advantage.
> >
> > ---
> >
> > >***What is the computational complexity (e.g., wall-clock time) of the HOLLM framework, specifically the KD-tree construction and region scoring, relative to the (presumably expensive) LLM sampling calls? How does this overhead scale as the number of evaluated points t increases?***
> >
> > The overhead of KD-tree construction and region scoring is asymptotically negligible compared to the $\mathcal{O}(t^2)$ attention mechanism cost or API latency of the $M$ sequential LLM calls. Since both HOLLM and the global LLM baseline sample the same number of candidates M at each iteration, HOLLM adds complexity only to the non-bottleneck parts of the pipeline, such as the partitioning and score computation. We have addressed this in the revised paper PDF (Appendix H, Table 7, and in the response to Reviewer Y2NU).
> >
> > | Component | Complexity (per iter.) | Relative Cost |
> > | :--- | :--- | :--- |
> > | **KD-Tree Construction** | $\mathcal{O}(t \log t)$ | Negligible |
> > | **Region Scoring** | $\mathcal{O}(L)$ | Negligible |
> > | **LLM Inference** | $\mathcal{O}(M t^2)$ | **Dominant** |
> >
> > *$t$: total points, $L$: number of leaves, $M$: candidates per iter.*
> >
> > ---
> >
> > >***The paper's motivation (Fig. 1) suggests LLMs are bad at sampling (i.e., biased). The method's success implies this bias is now a useful "prior". Could you elaborate on what specific, useful properties this "meta-prior" is assumed to have, especially when constrained to small subregions?***
> >
> > We apologize for not making this clearer. The "meta-prior" we refer to describes the inductive structure that LLMs acquire during pretraining, i.e., an ability to infer functional relationships, smoothness, curvature, and constraints from a small set of in-context examples. Prior work (von Oswald et al., 2023) shows that Transformers perform *in-context gradient descent*: when conditioned on a sequence of $(x, f(x))$ pairs, the model internally adapts representations to fit the local geometry of the function rather than simply doing pattern matching. However, this capability is difficult for the LLM to express when sampling over a global, high-dimensional domain: global variability, irrelevant structure, and broad ranges of scale dilute the signal contained in the high-performing points. This is exactly the failure mode illustrated in Fig. 1, where global LLM sampling becomes highly biased. By restricting the model to smaller regions of the search space, HOLLM allows the LLM to capture local smoothness, curvature, and trends far more effectively than under globally scattered conditioning points. The LLM can easily interpret the axis-aligned bounds for each subregion and prevents drifting behavior or bias that may occur when the model samples across the full domain. In our new diversity analysis (Appendix E) the global LLM baseline rapidly collapses toward previously sampled points, while HOLLM maintains substantially higher proposal diversity across iterations, supporting our hypothesis.
> >
> > ---
> >
> > Thank you again for your valuable feedback and constructive criticism, which have significantly strengthened our paper and its narrative. We believe HOLLM represents a novel and practical contribution to efficient LLM-based global optimization. Given the comprehensive rebuttal, the new empirical evidence supporting the synergistic role of the LLM and partitioning framework, and the positive feedback from several reviewers, we respectfully request that you consider raising your score to reflect the substantial improvements made to the submission.
> >
> > ---
> >
> > *-- References --*
> >
> > von Oswald et al. "Transformers learn in-context by gradient descent". In ICML 2023

---

### Official Review · Reviewer_Uiwf · 2025-11-01

**Soundness:** 3
**Presentation:** 3
**Contribution:** 3
**Rating:** 6
**Confidence:** 4

**Summary:**

The authors propose a method called HOLLM that partition the space of search using KD-Tree and then using a bandit-style algorithm that select from which part of the space sample new candidates points using LLMs. The motivation of the method bases on the observation that LLMs are not good by covering the space optimally. The authors show extensive results, including a comparison with a vanilla LLM implementation.

**Strengths:**

- Authors showcase extensive results demonstrating the applicability of HOLLM.
- The authors provide ablation studies of the design choices on the appendix.
- In general the paper is well written and presented.

**Weaknesses:**

- My only doubt is that the authors develop a method based on the main motivation that LLMs are not efficient covering the space of solution. However, it seems the method not necessarilly needs an LLM in its design. On this point why LLMs are revelant for this method? My first assumption would be because LLMs have strong inductive bias about the problem and make them sample more efficiently. However, the inductive biases would be given by the context that is given to the LLM. This is a point that is not properly discussed in the paper. Can you elaborate on this point?
- If HOLLM actually needs a LLM to perform well then it seems that is necessary a more extensive comparison between HOLLM and the LLM baseline to understand better why the first perform better than the second. Some intuition is given as motivation, but I guessed some metrics can be tracked to prove this point with the actual experiments.

**Questions:**

See weaknesses

---

> ### Author Response · Authors · 2025-11-20
>
> Thank you for the constructive feedback. Below we address your two main concerns regarding (1) why HOLLM actually needs an LLM, and (2) whether we can provide deeper empirical evidence comparing HOLLM to the vanilla LLM baseline.
>
> ---
>
> >***Why LLMs Are Relevant / Necessary for HOLLM***
>
> We want to emphasize that we do not provide any contextual information related to the tasks, except the inputs. For instance, text such as “You are given task X”, where X can be “Penicillin production”, are not present in our prompts (see the prompt examples in Appendix D as well). HOLLM and the LLM baseline only have access to the numerical $(x, f(x))$ values. Differently from the global LLM baseline or approaches such as Llambo, HOLLM is explicitly designed to leverage the inductive bias of LLMs in a localized way. We believe that when LLMs are used as in-context learners, as in our case, they leverage the specific ability of Transformers to perform in-context gradient descent (demonstrated in von Oswald et al. 2023), which becomes most effective when localized.
>
> - **LLMs carry a meta-prior about functional structure.** As shown in our motivating experiment, even though LLMs may encode strong priors, they struggle to express them when forced to sample from a global, high-dimensional domain. Localizing the sampling task through partitioning lets the LLM express this inductive bias much more effectively. Moreover, recent theoretical work (e.g., von Oswald et al., 2023) demonstrates that Transformers implement gradient descent updates within their forward pass. When conditioned on observed $(x, f(x))$ pairs, the LLM does not just match patterns; it adapts its internal representation to locally fit the function geometry (curvature, smoothness).
> - **Partitioning reduces generation difficulty.** The failure mode of a global LLM sampler is due to poor coverage and high clustering which might be affected by the global noise in the samples. By restricting the LLM to a small subregion (a leaf in our hierarchy), we simplify the optimization landscape. The LLM only needs to extrapolate local structure rather than global variability. We also empirically investigate this by computing the diversity in the generated samples at each iteration from HOLLM and the global LLM. We have added this diversity analysis in Appendix E and as described below, HOLLM maintains higher exploratory behaviour throughout iterations, as compared to the global LLM that collapses to already evaluated points.
> - **Replacing the LLM with uniform sampling and local GP-EI.** A non-LLM sampler inside the partitions (e.g., Uniform, Latin Hypercube) lacks this learned optimization capability. As we show in our experiments (Figures 3 to 5 in main paper and Figure 16 in the appendix), replacing the LLM with random sampling (the RS + KD-Tree baseline) or GP-EI degrades performance significantly, confirming that HOLLM’s success depends not only on the KD-tree partitioning, but also on the LLM predictive and sampling capabilities.
>
> ---
>
> >***Additional Empirical Comparison to Strengthen the Claim***
>
> We agree that a deeper comparison against the vanilla LLM baseline will strengthen the paper. We have conducted new experiments analyzing the diversity of generated samples relative to the in-context examples. In the revised PDF version, we have added a diversity analysis in Appendix E. To measure the diversity in generated samples, we introduce the *ICL Divergence* ($D_{ICL}$) metric, which measures the average distance between new proposals and their nearest in-context example. Results with this metric on the real-world benchmarks (Figure 16) show that the vanilla LLM’s $D_{ICL}$ drops rapidly across trials (approaching zero), indicating severe mode collapse where the model simply mimics existing high-performing points, without any exploration. In contrast, HOLLM consistently maintains a significantly higher $D_{ICL}$, proving that the partitioning helps the LLM explore new regions while still leveraging its meta-prior. Following the ICL implicit gradient descent dynamics as explained above, our algorithm allows the LLM to apply its gradient-descent-like capabilities locally, but prevents it from collapsing into a global trivial solution. By avoiding redundant resampling of known optima while still exploiting the LLM's structural understanding, HOLLM becomes a more sample efficient alternative to the global LLM optimizer.
>
> We sincerely thank the reviewer for their constructive criticism. The discussion and the new diversity metrics have significantly strengthened the paper’s narrative. We are happy to engage in further discussion in case you have additional concerns. We would also appreciate it very much if you would consider raising your score.
>
> ---
>
> *-- References --*
>
> von Oswald et al. "Transformers learn in-context by gradient descent". In ICML 2023

---

### Author Response · Authors · 2025-11-20
**General Response to All Reviewers**

We sincerely thank all reviewers for their thoughtful and constructive feedback. Your detailed critiques and suggestions played a significant role in strengthening both the empirical evidence and the narrative of our paper. We are pleased that the reviewers recognized the core strengths of our work, namely, *the novelty aspect* (reviewers Uiwf, Y2NU, LeUX), *our core motivation* (reviewer dF12) and *the extensive empirical evaluations across diverse tasks* (reviewer Uiwf, LeUX). In response to the primary action items from the reviewers, we performed several crucial new experiments during this discussion phase, which we show in the **new Appendix E-J of the revised paper PDF (4.5 new added pages at the end of the file, with red-colored text)**. Due to the page constraint, we kept these additional experiments in the appendix, however, we will integrate the most important ones in the main paper upon acceptance. The table below summarizes these experiments.

| Reviewer Action Item | Experiment Performed & Key Finding | Revised PDF Addition |
| :--- | :--- | :--- |
| **Deeper HOLLM vs. Global LLM comparison (Uiwf)** | Introduced a metric to quantify LLM sampling diversity. Results show that the global LLM rapidly suffers from *mode collapse*, while HOLLM maintains high exploratory behavior throughout. | Appendix E, Figure 16 |
| **LLM vs. Simpler Samplers (dF12)** | HOLLM significantly outperformed KD-Tree + Random Search (RS) and KD-Tree + GP-EI, confirming the benefit of the localized LLM candidate sampler. | Appendix F, Figure 18 |
| **Ablating Scoring Function Components (Y2NU)** | We evaluated the importance of the full composite scoring rule (including the geometric term and UCB-V); simpler variants like UCB1 led to worse performance. | Appendix F, Figure 17 |
| **Multi-LLM Ablation (LeUX)** | We expanded the multi-LLM comparison in Figure 8 to another continuous and discrete task. We showed HOLLM's benefit is more prominent when using smaller open-weights LLMs. | Appendix I, Figure 20 |
| **Surrogate Prediction Performance (Y2NU)** | We compared the LLM surrogate against Gaussian Processes (GPs) and TabPFN. The LLM demonstrated superior few-shot calibration and ranking accuracy on real-world tasks. | Appendix G, Table 6 |
| **Performance under Observation Noise (Y2NU)** | We tested HOLLM on Vehicle Safety with noise injected in the function values. HOLLM shows robustness and the performance gap towards the global LLM baseline increases. | Appendix J, Figure 21 |
| **Computational Complexity (dF12, Y2NU)** | We provided a formal Big-O analysis, concluding that the overhead coming from partitioning and region selection is asymptotically negligible compared to the dominant $\mathcal{O}(t^2)$ LLM inference cost. | Appendix H, Table 7 |

---

These extensive additions comprehensively address the reviewers' major concerns, *providing strong empirical and analytical justification* for combining the LLM's with our adaptive partitioning framework. We believe the paper is significantly stronger due to your insightful guidance and are happy to engage further in discussion if you have additional concerns. We respectfully request that you consider raising your score to reflect the substantial improvements made to the submission. Thank you again for your time and expertise.

---

### Meta-Review · Area_Chair_oRK9 · 2026-01-02

**Summary:**

This paper introduces HOLLM, a novel black-box optimization algorithm that combines adaptive KD-tree-based space partitioning, multi-arm bandit-style partition selection (using a custom scoring rule), and localized LLM-based candidate proposal. The authors rigorously benchmark HOLLM on diverse synthetic and real-world tasks, compare it against state-of-the-art optimization and LLM-based baselines, and conduct extensive ablation and sensitivity studies. The revised submission features significant new experiments and clarifies the roles of various methodological components.

Despite some initial reservations regarding novelty and the incremental contribution over standard hierarchical optimization frameworks, the authors have convincingly demonstrated—through new ablations, robustness tests, and a careful response to critique—that HOLLM is an empirically strong and pragmatically valuable algorithm. The paper is well-positioned to inform and inspire both researchers developing LLM-based optimizers and practitioners seeking robust, resource-efficient black-box optimization solutions.

**Reviewer Concerns:**

Reviewers' main concerns are solved.

**Reviewer Scores:**

One reviewer has admitted to updating his score.

---

### Decision · Program_Chairs · 2026-01-26

Accept (Poster)